



# Two-way feedback mechanism between unfavorable meteorological conditions and cumulative aerosol pollution exists in various haze regions of China

Junting Zhong[1], Xiaoye Zhang[1,2*], Yaqiang Wang[1], Jizhi Wang[1], Xiaojing Shen[1], Hongsheng Zhang[3], Tijian Wang[4], Zhouqing Xie[2,5,6], Cheng Liu[2,5,6], Hengde Zhang[7], Tianliang Zhao[8], Junying Sun[1], Shaojia Fan[9], Zhiqiu Gao[8], Yubin Li[8], Linlin Wang[10]

[1]Chinese Academy of Meteorological Sciences, China Meteorological Administration, Beijing, 100081, China.

[2]Center for Excellence in Regional Atmospheric Environment, IUE, Chinese Academy of Sciences, Xiamen, 361021, China.

[3]Laboratory for Climate and Ocean-Atmosphere Studies, Department of Atmospheric and Oceanic Sciences, School of Physics, Peking University, Beijing, 100081, China

[4]School of Atmospheric Sciences, Nanjing University, Nanjing, 210023, China

[5]Key Lab of Environmental Optics and Technology, Anhui Institute of Optics and Fine Mechanics, Chinese Academy of Sciences, Hefei, 230031, China

[6]School of Earth and Space Sciences, University of Science and Technology of China, Hefei, 230026, China

[7]National Meteorological Center, China Meteorological Administration, Beijing, 100081, China

[8]Nanjing University of Information Science & Technology, Nanjing, 210044, China

[9]Sun Yat-sen University, Guangzhou, 510275, China

[10]State Key Laboratory of Atmospheric Boundary Layer Physics and Atmospheric Chemistry, Institute of Atmospheric Physics, Chinese Academy of Sciences, Beijing 100029, China

*Correspondence to*: Xiaoye Zhang (xiaoye@cma.gov.cn)

**Abstract.** Accompanied by unfavorable meteorological conditions with stable stratification in various haze regions of China, persistent heavy aerosol pollution episodes lasting more than 3 consecutive days (HPEs) frequently occur, particularly in winter. In the North China Plain (NCP), explosive growth in $PM_{2.5}$, which occurs in some HPEs, is dominated by a two-way feedback mechanism between further worsened unfavorable meteorological conditions and cumulative aerosol pollution. However, whether such a two-way feedback mechanism exists in other key haze regions is uncertain; these regions include the Guanzhong Plain (GZP), the Yangtze River Delta (YRD) region, the Two Lakes Basin (TLB), the Pearl River Delta (PRD) region, the Sichuan Basin (SB), and the Northeast China Plain (NeCP). In this



study, using surface PM₂.₅ and radiation observations, radiosonde observations, and reanalysis data, we
observed the existence of a two-way feedback mechanism in the above six regions. In the SB, this two-
way feedback mechanism is weak due to the suppression of cloudy mid-upper layers. In the more polluted
NCP, the FWRP, and the NeCP, the feedback is more striking than that in the YRD, the TLB, and the
PRD. In these regions, the feedback of worsened meteorological conditions on PM₂.₅ explains 60~70%
of the increase in PM₂.₅ during the cumulative stages (CSs). For each region, the low-level cooling bias
becomes increasingly substantial with aggravating aerosol pollution and a closer distance to the ground
surface. With PM₂.₅ mass concentrations greater than 400 μg m⁻³, the near-ground bias exceeded -4 ºC in
Beijing and reached up to approximately -4 ºC in Xi'an; this result was caused by accumulated aerosol
mass to some extent. In addition to the increase in PM₂.₅ caused by the two-way feedback, these regions
also suffer from the regional transport of pollutants, including inter-regional transport in the FWRP,
trans-regional transport from the NCP to the YRD and the TLB, and southwesterly transport in the NeCP.
**1 Introduction:**

In China, 94% of the total population is distributed in eastern China (Yang et al., 2016), in which

aerosol pollution has drawn wide attention. In the basins and plains in eastern China, aerosol pollution
episodes frequently occur in winter, and these episodes cause economic loss and have adverse effects on
human health (Chen et al., 2013;Bai et al., 2007;Matus et al., 2012). For example, in January 2013,
persistent heavy aerosol episodes affected 600 million people over a span of 1.4 million square kilometers
(http://www.infzm.com/content/95493), which led to hundreds of flight cancellations and an increase in
the number of reported respiratory disease cases (Ji et al., 2014). During the wintertime (i.e., Jan., Feb.,
and Dec.) from 2013 to 2017, more than 28 persistent heavy aerosol pollution episodes that lasted for
more than 3 consecutive days (HPEs) occurred in Beijing; the peak value of particulate matter smaller
than 2.5 μm in diameter (PM₂.₅) ranged from ~200 μg m⁻³ to ~ 800 μg m⁻³, with a mean duration longer
than 5 days (Zhong et al., 2018a) & (Zhong et al, Tellus B, accepted). The main cause of frequent
pollution episodes is the massive emissions of air pollutants produced by intense living and industrial
activities in the basins and plains (Zhang et al., 2009a;Zhang et al., 2013;Zhang et al., 2012a). In addition
to pollutant emissions, the relatively closed terrains of basins and plains limit the diffusion of aerosols
and their precursors to the surrounding areas (Su et al., 2004;Zhu et al., 2018). Under stable





meteorological conditions, aerosol pollution forms and further accumulates (Zhang et al., 2013;Zhong et
al., 2017).
In winter, unfavorable meteorological conditions for pollution dispersion that generally have strong
static stability lead to aerosol pollution, and after accumulating to some extent, aerosols will change the
atmospheric structure by interacting with solar radiation (Zhong et al., 2018b;Boucher et al., 2013). The
dominant scattering aerosol will back-scatter solar radiation, causing a reduction in the amount of solar
radiation that reaches the surface, which causes a cooling effect through atmospheric circulation and
vertical mixing. After analyzing the causes of HPEs in Beijing, previous studies found that elevated $PM_{2.5}$
mass (to a certain extent) scattered more solar radiation to space, which substantially reduced surface
radiation (i.e., the cumulative sum of hourly radiant exposure reduced by 89% and 56%, respectively,
from clean stages to CSs) and subsequently reduced surface temperature under slight or calm winds
(Zhong et al., 2018b;Zhong et al., 2017). The temperature reduction induces or reinforces an inversion
that further weakens turbulence diffusion and results in a lower boundary layer (BL) height. This
feedback effect of further worsened meteorological conditions aggravates $PM_{2.5}$ pollution and forms a
two-way feedback mechanism between unfavorable weather conditions and cumulative $PM_{2.5}$ pollution
(Zhong et al., 2017); this condition also decreases the near-ground saturation vapor pressure to increase
the relative humidity (RH), which will further enhances aerosol hygroscopic growth and accelerates
liquid-phase and heterogeneous reactions (Ervens et al., 2011;Pilinis et al., 1989;Kuang et al.,
2016;Zhong et al., 2018b;Zhong et al., 2018a). The mutual promotion mechanism between unfavorable
meteorological conditions and cumulative aerosol pollution also appeared in other cities in the North
China Plain, including Tangshan, Xingtai, Zhengzhou and Nanyang (Liu et al, 2018, under review).
Whether the two-way feedback mechanism exists in other basins and plains in eastern China, which
have weaker aerosol pollution than that in the North China Plain, is unclear. If such a feedback exists, its
magnitude requires further investigation. Currently, to the best of our knowledge, studies on the existence,
magnitude and comparison of the two-way feedback in these basins and plains are insufficient. Overall,
we lack a comprehensive understanding of the feedback mechanism in China. Therefore, here we used
surface $PM_{2.5}$ mass concentrations, radiosonde observations of meteorological factors, meteorological
index parameter-linking aerosol pollution and meteorological factors (PLAM), and ERA-interim
reanalysis data from the European Center for Medium Weather Forecasting (ECMWF) to investigate the



89 two-way feedback mechanism in the key regions of populous eastern China (Yang et al., 2016), including

90 the Guanzhong Plain, the Yangtze River Delta, the Two Lakes Basin, the Pearl River Delta, the Sichuan

91 Basin, and the Northeast China Plain, which are densely populated and economically developed regions

92 that include massive industrial pollution sources, agricultural pollution sources, motor vehicle pollution

93 sources and domestic pollution sources. In the above regions, heavy aerosol episodes often occur in the

94 regional central cities with denser populations and stronger pollutant emissions, including Xi'an, Nanjing,

95 Shanghai, Wuhan, Guangzhou, Chengdu, and Shenyang. In the above cities, the impact of aerosol

96 pollution episodes on the economy, society and health is far-reaching. Therefore, we focus on the

97 feedback mechanism in the above cities to represent the overall conditions in the five major haze regions

98 of China, namely, II) the North China Plain (also called Hua Bei Plain) in N. China, plus the Guanzhong

99 Plain; III) E. China with the main body in the Yangtze River Delta area; V) S. China with the most areas

100 of Guangdong and the Pearl River Delta area; IV) The Sichuan Basin in S. W. China, and I) Northeast

101 China Plain (Zhang et al., 2012b) (Fig. 1). In addition, due to the lack of meteorological radiosonde

102 observations in Guangzhou, we supplemented related observations in its adjacent city, Qingyuan.

## 2 Materials and methods:

### 2.1 PM$_{2.5}$ observations:

105 Since January 2013, the Ministry of Environmental Protection has been monitoring the PM$_{2.5}$ mass

106 concentrations in real time at over 1000 environmental monitoring stations established in different

107 regions of China. In this study, we used the hourly PM$_{2.5}$ mass concentrations provided by the Ministry

108 of Environmental Protection from December 1, 2016 to January 10, 2017 and the respective averaged

109 PM$_{2.5}$ mass concentrations of all the urban stations in Xi'an, Yuncheng, Shenyang, Chengdu, Wuhan,

110 Nanjing, Shanghai, Jinan, Guangzhou and Qingyuan.

### 2.2 Meteorological radiosonde observations:

112 In China, 120 stations have been observing vertical meteorological factors using L-band sounding

113 radars. Their accurately positioned radar systems collect reliable meteorological data each second; thus,

114 these data have high spatial and temporal resolutions (Tao, 2006). In this study, we used the L-band

115 sounding radar data from the meteorological stations in Xi'an, Shenyang, Chengdu, Wuhan, Nanjing,





Shanghai and Qingyuan; these stations observed several meteorological factors, including wind,
temperature and RH, twice each day at 0800 (BJT) and 2000 (BJT) from December 1, 2016 to January
10, 2017. The meteorological factors were analyzed in detail below the height of 3 km. The heights from
the surface to 1 km, from 1 km to 2 km, and from 2 km to 3 km are termed the low-level, mid-level, and
upper-level heights, respectively.
**2.3  Surface meteorological data:**
Since 2001, national weather stations have been conducting hourly automatic observations. Some
of the stations began to record observations every 5 and 10 minutes starting in 2011. This study used the
hourly meteorological observation data, including temperature, pressure, RH, wind and visibility at the
National Automatic Weather Stations (AWS) provided by the National Meteorological Information
Center of China Meteorological Administration (NMICMA). The time range of the selected data is from
December 1, 2016 to January 10, 2017.
We also used an hourly radiant exposure data set of national meteorological radiation factors (V2.0)
provided by the NMICMA. This dataset contains 104 radiation stations, including primary stations with
global, direct, scattered, reflected, and net radiation, secondary stations with global and net radiation, and
tertiary stations with only global radiation. These radiation stations recorded hourly basic radiant
exposure data and the corresponding station information (i.e., latitude, longitude and altitude) starting in
1993. In this study, we used the global, direct and net radiant exposure from December 1, 2016 to January

10, 2017.

**2.4  PLAM data**
Based on the definition and calculation formula of a parameter that links aerosol pollution and
meteorological factors (PLAM) (Wang et al., 2013;Zhang et al., 2015;Zhang et al., 2009b;Wang et
al., 2012), we obtained the PLAMs in Xi'an, Nanjing, Wuhan, Qingyuan, Chengdu, and Shenyang
using surface meteorological factors. PLAM includes two major separate factors: (1) the initial
meteorological conditions $\alpha(m)$ associated with the atmospheric condensation processes and (2) a
dynamic effective parameter associated with the initial contribution of air pollution $\beta(c')$:
$$PLAM = \alpha(m) \times \beta'(c). \qquad (1)$$
This calculation mainly indicates the regional atmospheric stability and the air condensation



ability. The details of the calculation have been presented in previous studies (Wang et al.,
2013;Wang et al., 2012).

**2.5 ECMWF ERA-Interim data**

ERA-Interim is ECMWF's latest global atmospheric reanalysis, which extends back to 1979 and
continuously updates in real time (Dee et al., 2011). It is produced with a 4-dimensional variational data
assimilation scheme and advances forward in time using 12-hour analysis cycles (Dee et al.,
2011;Thépaut et al., 1996). Before assimilation, all data are subject to gross, redundancy and background
quality controls, which resulted in a large drop between the total number of available data and the number
of data used in the assimilation. The mean daily usage rate of radiosondes is no more than 50% over the
entire time period (Poli et al., 2010). In addition, although the effect of aerosols on radiative transfer in
the atmosphere is modeled based on prescribed climatological aerosol distributions (Dee et al., 2011), it
has not been considered to be the two-way feedback mechanism between the cumulated aerosol pollution
and the worsened meteorological conditions (Simmons, 2006). Therefore, the magnitude of the feedback
mechanism could be statistically reflected by the gaps between the ERA-interim reanalysis and the
meteorological radiosonde observations. The disparities have been used to present the observational
evidence of aerosol-PBL interactions in Beijing (Ding et al., 2016;Huang et al., 2018).
In this study, we used ERA-Interim data with a horizontal resolution of 0.125° × 0.125°. Its
mandatory pressure levels include 1000, 975, 950, 925, 900, 875, 850, 825, 800, 775, 750, and 700 hPa.
According to these pressure layers, we interpolated the radiosonde observations and calculated the
vertical temperature differences between the ERA-interim reanalysis and the interpolated sounding data
at 20:00 (BJT).

**3 Results and Discussions:**

Based on the consistent variation trends in visibility, China is classified into nine typical regions
(Zhang et al., 2012b). Five of these regions have experienced striking declines in visibility in recent
decades, including (1) the North China Plain and the Guanzhong Plain in North China; (2) the Yangtze
River Delta region and the Two Lakes Basin along the middle and lower reaches of the Yangtze River;
(3) the Pearl River Delta region in South China; (4) the Sichuan Basin in Southwest China; (5) and the
Northeast China Plain (Fig. 1). The areas with declined visibility coincide with the basins and plains in



eastern China because these basins and plains are densely populated and topographically enclosed;
additionally, these areas emit and produce massive air pollutants, including primary aerosols and
secondary aerosols from gas-to-particle conversion. These aerosols locally accumulate to continuously
reduce visibility. By comparing the mean $PM_{2.5}$ mass concentration from December 1, 2016 to that of
January 10, 2017 in the five regions that experienced declines in visibility (Fig. 2), we found that the
heaviest aerosol pollution occurred in the North China Plain, and it was followed by the Guanzhong Plain.
The areas with the next highest aerosol pollution were the Sichuan Basin and the Northeast China Plain.
The Two Lakes Basin and the Yangtze River Delta experienced less aerosol pollution. Finally, the Pearl
River Delta had the least aerosol pollution.
**3.1  Striking two-way feedback mechanism of the polluted Guanzhong Plain with inter-regional**
**pollution transport.**
To the north of the Loess Plateau and the south of the Qinling Mountains, the Guanzhong Plain has
a narrow and closed terrain (Fig. 1), and its climatic and meteorological conditions are distinctive from
those of the surrounding areas. Compared with the plateau to the north, the Guanzhong Plain is less
affected by northerly cold and clean winds, and these conditions favor the accumulation of pollutants.
However, because the Loess Plateau is lower in elevation than the Hengduan Mountains and the Daba
Mountains located to the northwest of the Sichuan Basin, the barrier effect of the plateau on the northerly
cold air is weaker than that of those mountains (Fig. 3 (b) and Fig. 10 (b)). Because the North China
Plain is bordered to the west by the Taihang Mountains and the Lvliang Mountains (Fig. 1), the
Guanzhong Plain is rarely affected by pollutant transport from the North China Plain; however, air
pollution is highly correlated among the different cities in the Guanzhong Plain. To the west of this plain,
Xi'an lies north of the Wei River and the Loess Plateau and south of the Qinling Mountains (Fig. 1). Due
to its enclosed topography, Xi'an frequently experiences heavy urban aerosol pollution.
From December 1, 2016 to January 10, 2017, two HPEs appeared in Xi'an and persisted for more
than 7 days with peak mass concentrations greater than 400 μg m$^{-3}$ (Fig. 3 (a), dark blue lines). During
HPE$_{1-2}$, we observed a striking two-way feedback mechanism between the worsened weather conditions
and the cumulated aerosol pollution (Fig. 3, red and white boxes). When the near-ground $PM_{2.5}$
accumulates to a certain extent, the particles scatter more solar radiation back to space, which
substantially reduces the surface radiation (Fig. 3 (e), red boxes) and consequently lowers the near-



surface temperature (Fig. 3 (c), white boxes). Under slight or calm winds (Fig. 3 (b), red boxes), the
temperature reduction induces or reinforces inversions, which further weaken turbulence diffusion to
suppress the diffusion of water vapor and pollutants (Zhong et al., 2017;Zhong et al., 2018a); these
conditions also decrease the near-ground saturation vapor pressure to increase the RH (Fig. 3 (d), red
boxes), which further enhances aerosol hygroscopic growth and accelerates liquid-phase and
heterogeneous reactions (Cheng et al., 2016;Fang et al., 2016;Tie et al., 2017). This type of two-way
feedback mechanism leads to worsened meteorological conditions and elevated $PM_{2.5}$ mass
concentrations.
During $HPE_{1-2}$, we also observed an increase in the $PM_{2.5}$ mass concentration caused by pollutant
transport. The aerosol pollution in Xi'an might be aggravated by the transport of pollutants from the
eastern polluted plain area with heavily polluted cities, including Yuncheng and Linfen. To reveal the
effects of air pollutant transport from the eastern plain on the aerosol pollution in Xi'an, we compared
the variation trends in the $PM_{2.5}$ mass concentrations in Xi'an and Yuncheng under lower northwesterly
winds (Fig. 3 (a, b)). We found that during TSs in Fig. 3 (orange boxes), low-level northwesterly winds
would transport pollutants below the BL to maintain or aggravate the aerosol pollution in Xi'an when
Yuncheng is heavily polluted; however, when Yuncheng had good air quality, the aerosol pollution in
Xi'an was lighter or even eliminated.
In addition to the scavenging effect of clean northwesterly winds on aerosol pollution, pollution
elimination mainly depends on lower strong northwesterly winds and mid-upper level southerly winds.
Because the Loess Plateau north of Xi'an is sparsely populated with rare air pollutant emissions, lower
strong and clean northwesterly winds would blow away aerosol pollutants in Xi'an, causing a subsequent
rapid improvement in the air quality (Fig. 3 (a, b)). Since the mid-upper level southerly winds transport
water vapor to Xi'an from the area south of Xi'an, the mid-upper (or whole-layer) RH level is
considerably enhanced (i.e., greater than 96%) (Fig. 3 (b, d), brown boxes), which causes the $PM_{2.5}$ to
enter the fog-cloud phase and possibly produces precipitation that eliminates pollutants through wet
removal (Fig. 4 (d), blue dots represent precipitation).
**3.2  Affected by trans-regional pollution transport from the North China Plain, the Yangtze River**
**Delta region subsequently experiences the two-way feedback, where the clearing of pollution**
**depends on persistent stronger northerly winds, or southeasterly warm, humid winds through**
**fog-cloud conversion and wet removal.**



Located in the lower reaches of the Yangtze River, the Yangtze River Delta is a triangle-shaped
metropolitan region. It covers an area of 211,700 km$^{-2}$ and was home to more than 150 million people as
of 2014 (http://www.ndrc.gov.cn/zcfb/zcfbghwb/201606/t20160603_806390.html). The urban build-up
in this area has given rise to what may be the largest concentration of adjacent metropolitan areas in the
world. The Yangtze River Delta has a marine monsoon subtropical climate with cool and dry winters.
Situated in the Yangtze River Delta, Nanjing is the second largest city in the East China region. The south,
north, and east sides of the city are surrounded by the Ningzheng Ridges (Fig. 1), while the Yangtze River
flows along the west and part of the north sides.
From December 1, 2016 to January 10, 2017, four aerosol pollution episodes occurred in Nanjing
(Fig. 4 (a), blue boxes). One of these episodes lasted for less than 3 days and had light pollution, while
the other three episodes persisted for more than 5 days and had peak mass concentrations greater than
150 μg m$^{-3}$; thus, these three episodes are termed HPEs (Fig. 4 (a)). During these three HPEs, although
the PM$_{2.5}$ mass concentration was much lower than that in Beijing, the aerosol pollution formation was
similar to that in the latter, including earlier transport stages (TSs) and later cumulative stages (CSs)
(Zhong et al., 2017;Zhong et al., 2018a). During the TSs in the HPEs, strong northerly winds transport
aerosol pollutants from the polluted North China Plain to the Yangtze River Delta region below and over
the BL (i.e., long-distance pollution transport), which induces a striking increase in the PM$_{2.5}$ mass
concentration in Nanjing and a reduction in the PM$_{2.5}$ mass concentration in Jinan, a regional center city
representative of the pollution conditions in the NCP (Fig. 4 (a, b)). To some extent, based on the PM$_{2.5}$
mass, the two-way feedback mechanism is activated during the CSs, in which we observed surface
radiation reductions, near-surface inversions, low-layer RH enhancement, and increased PM$_{2.5}$ mass
concentrations under slight winds (Fig. 4). Due to the lighter aerosol pollution in Nanjing, the two-way
feedback mechanism is weaker than that in Beijing (Fig. S1, 4 (a)). In addition, the mechanism might be
weakened by relatively strong lower winds (compared to those in Beijing) (Fig. S1, 4 (b)), which are
unfavorable for the accumulation of aerosols.
To reveal the regional pollutant transport patterns from the North China Plain to the Yangtze River
Delta region, we calculated the concentration difference in the PM$_{2.5}$ mass between the start time and the
end time of the TSs in HPE$_{1,2,4}$ (Fig. 5). We found that the southern area of the North China Plain
experienced a substantial reduction in its PM$_{2.5}$ mass concentration, while an increase occurred in the



middle and lower reaches of the Yangtze River, including the Two Lakes Basin and the Yangtze River
Delta region; these results indicate the process of regional pollutant transport from the North China area
to the East China area under strong northwesterly winds. In the winter of 2017, we also observed this
pollution transport (Fig. 6, orange boxes), after which persistent northerly winds blew pollutants away
(Fig. 6, purple boxes),. In addition to the blowing effect of persistent northerly winds, eliminating
pollution in Nanjing mainly depends on strong southeasterly winds, which transport warm, humid, and
clean air from the Yellow Sea and the East China Sea; this air also blows the aerosol pollutants in Nanjing
away (Fig. 4 (b, c, d)). In addition, transported water vapor increases the RH (Fig. 4 (b, d)), which causes
the $PM_{2.5}$ to enter the fog-cloud phase and possibly produces precipitation that eliminate pollutants
through wet removal (Fig. 4 (d), blue dots represent precipitation).

Consistent with the results observed in Nanjing, Shanghai also experienced long-distance pollution

transport below and over the BL under northwesterly winds (Fig. 8 (a, b), orange boxes). After $PM_{2.5}$
accumulated to some extent, we observed a two-way feedback mechanism, including reduced radiation,
near-surface inversions, RH enhancement in the lower parts of BL, and increases in $PM_{2.5}$ mass
concentration under slight or calm winds (Fig. 8 (a, b, c, d, e) red and white boxes); however, the
magnitude of the feedback was weaker than that observed in Nanjing (Fig. 4). Because Shanghai is closer
to the sea than Nanjing, it is more susceptible to warm, humid southeasterly winds from the sea, which
carry more water vapor to Shanghai than to Nanjing (Figs. 4 & 8, (b, d)).

**3.3   The two-way feedback mechanism exists in the Two Lakes Basin. Aerosol pollution is also**
**worsened by the trans-regional pollution transport from the North China Plain and eliminated**
**by fog-cloud conversion and wet removal from mid-upper southwesterly winds.**

The Two Lakes Basin is in the middle reaches of the Yangtze River. With the Sichuan Basin

bordered to the northwest by the Daba Mountains (Fig. 1), the Two Lakes Basin is rarely affected by
pollutant transport from polluted cities in Sichuan Basin. The north side of the Two Lakes Basin is
connected to the North China Plain through the Suizhou Corridor and the Nanyang Basin (Fig. 1); thus,
the Two Lakes Basin is vulnerable to pollution transport from the North China Plain, which experiences
the heaviest aerosol pollution in China (Fig. 2). As a large exorheic basin surrounded by low ridges or
mountains, the Two Lakes Basin more frequently exchanges air masses with its surroundings, with wind
speeds much higher than those in Sichuan Basin. Situated in the eastern Two Lakes Basin, Wuhan is the





most populous city in Central China. The Yangtze and Han rivers wind through this city, which has a
southern hilly and middle flat terrain (Fig. 1).

From December 1, 2016 to January 10, 2017, four aerosol pollution episodes occurred in Wuhan

(Fig. 7 (a), blue boxes). Three of these episodes lasted longer than 5 days and had peak mass
concentrations greater than 150 μg m⁻³, which are termed HPEs (Fig. 8 (a)). During these three HPEs,
we observed a two-way feedback mechanism in the red boxes (Fig. 8), including surface radiation
reductions, near-surface inversions, low-level RH enhancement, and increases in $PM_{2.5}$ mass
concentrations under slight or calm winds (Fig. 4). Similar to the conditions observed in Nanjing, Wuhan
experienced lighter aerosol pollution than Beijing (Fig. S1, 7 (a)); thus, the two-way feedback mechanism
is weaker than that observed in Beijing.

Figure 5 shows the regional pollutant transport from the North China Plain to the Two Lakes Basin,

which also aggravates the $PM_{2.5}$ pollution in Wuhan. As shown in the orange boxes of Fig. 8, the lower
northerly winds transport pollutants from the north of Wuhan to below Wuhan and sometimes over the
BL, which results in increasing $PM_{2.5}$ mass concentrations. Therefore, favorable northerly winds
establish a pollution linkage between the North China Plain and the middle and lower reaches of the
Yangtze River (including the Yangtze River Delta and the Two Lakes Basin), which have low and flat
terrains (Fig. S2). However, if the northerly winds are persistent and strong enough, they will blow the
aerosol pollutants out of the North China Plain entirely and then transport clean and cold winds to Wuhan;
under these conditions, the $PM_{2.5}$ mass concentration first increases and then decreases dramatically. This
phenomenon was observed from December 12 to 14, 2016 and is shown in Fig. 8.

In addition to the blowing effect of the strong, persistent northerly winds, clearing the pollution in

Wuhan mainly depends on the mid-upper level southerly winds, particularly the southwesterly winds,
which transport water vapor to Wuhan from the south, substantially enhancing the RH (over 96%) (Fig.
8 (b, d), brown boxes); these conditions cause the $PM_{2.5}$ to enter the fog-cloud phase and often produce
precipitation that eliminates pollutants through wet removal (Fig. 8 (d), blue dots represent precipitation).
**3.4   The two-way feedback mechanism also exists in the less polluted Pearl River Delta region.**

**This area is also humidified by upper southerly winds from the South China Sea and is purified**

**by lower clean, cold northeasterly winds from the northern mountains.**

Located in the southeastern area of Guangdong Province, the Pearl River Delta is one of the most



populous and densely urbanized regions in the world. This low-lying area is surrounded by the Pearl
River estuary, where the East River, West River, and North River converge to flow into the South China
Sea. With the South China Sea to its south, the Pearl River Delta region is often influenced by southerly
sea winds; however, with the mountainous area in northern Guangdong to the north (Fig. 1), the Pearl
River Delta region is less affected by northerly cold and clean winds. Situated at the heart of the Pearl
River Delta region (Fig. 1), Guangzhou is the most populous city of Guangdong Province. However, due
to the lack of a meteorological radiosonde station in Guangzhou, we used the sounding observations from
Qingyuan, a city with similar $PM_{2.5}$ variation trends (Fig. 9 (a)); Qingyuan is located approximately 60
km to the north of Guangzhou.

From December 1, 2016 to January 10, 2017, the $PM_{2.5}$ mass concentration in Guangzhou and

Qingyuan is ~50 μg m$^{-3}$, which is much lower than that in Xi'an, Nanjing, Wuhan, Chengdu, and
Shenyang (Figs. 3, 4, 8, 7, 9, 12 (a)). During this period, only one HPE occurred, and it lasted for more
than 8 days with a peak mass concentration of approximately 150 μg m$^{-3}$ (Fig. 9, blue line). During this
episode, we observed surface radiation reductions, near-surface inversions, low-level RH enhancement,
and increases in the $PM_{2.5}$ mass concentration under slight or calm winds (Fig. 9, red/white boxes below
the blue line), which suggest that a two-way feedback mechanism exists in the region. Except for this
episode, we found that the $PM_{2.5}$ mass concentration increased during slight or calm winds but was still
below the threshold (Fig. 8, the red boxes before Jan 1, 2017) (Zhong et al, Tellus B, 2018, accepted);
thus, no inversion or increased RH occurred because the two-way feedback mechanism was not
effectively activated.

Clearing pollution from Qingyuan depends on the lower strong northeasterly winds, which transport

dry, cold, and clean air to decrease temperature and RH and blow aerosol pollutants away from Qingyuan.
(Fig. 9 (b, d), purple boxes). In addition to the blowing effect of the cold northeasterly winds, the aerosol
pollution in Qingyuan is also affected by the mid-upper level sea flows, which enhance the atmospheric
RH to cause the $PM_{2.5}$ to enter the fog-cloud phase and possibly produce precipitation that eliminates
pollutants through wet removal (Fig. 9 (d), blue dots represent precipitation).
**3.5  The two-way feedback mechanism is weakened by cloudy mid-upper layers in the humid**
**Sichuan Basin with aerosols accumulated under slight or calm winds. This area is capped by**
**upper-level temperature inversions caused by a layer of air moving east across the Tibet Plateau.**





Located in the upper reaches of the Yangtze River in southwestern China, the Sichuan Basin is a
lowland region surrounded by mountains on all sides (Fig. 1). Abutting the eastern edge of the Tibetan
Plateau to the west and northwest and the Daba Mountains and the Wu Mountains to the east and
northeast, respectively (Fig. 1), the Sichuan Basin is rarely affected by cold northerly winds, which are
blocked by the high mountains. On the southern and southeastern sides, the Sichuan Basin is flanked by
the lower Yungui Plateau (Fig. 1), which is frequently affected by warm, humid southwesterly and
southeasterly airflows from the Bay of Bengal and the southeastern sea. Transported water vapor from
the south is blocked by the tall northern mountains and then accumulates in the Sichuan Basin. Located
at the western edge of the Sichuan Basin, Chengdu is surrounded by the highlands to the south, the high
and steep Longmen Mountains to the northwest, the Qionglai Mountains to the west, and the low
Longquan Mountains to the east. The enclosed topographical features lead to a lower wind speed and a
higher RH in Chengdu than in other parts of the Sichuan Basin.
From December 1, 2016 to January 10, 2017, three HPEs appeared in Chengdu (Fig. 10, blue boxes),
and these episodes lasted for more than 10 days and had peak mass concentrations greater than 200 μg
$m^{-3}$ (Fig. 10 (a)). During these three episodes, we observed thick mid-upper level fog/clouds above
Chengdu (Fig. 10 (d)), which was blocked by the surrounding mountains and upper-level inversions. The
mid-upper level cloud competes with the near-surface aerosols for solar radiation, i.e., as more solar
radiation is reflected by the mid-upper layer cloud, the near-surface aerosols receive less solar radiation.
Therefore, with cloudy mid-upper layers, more solar radiation is reflected back to cool the atmosphere
below the clouds, and this condition suppresses the two-way feedback mechanism between the
unfavorable weather conditions and the near-surface aerosols. Consequently, the two-way feedback was
weak and nearly no near-ground temperature inversion was observed (Fig. 10 (c)). Despite the lack of a
two-way feedback mechanism to aggravate aerosol pollution, the increase in the $PM_{2.5}$ mass
concentration is still under stable stratification dominated by slight or calm winds (Fig. 10, red boxes).
Comparing the RH variations in the two process of increasing $PM_{2.5}$ (Fig. 10 red boxes) during the HPE
from December 26, 2016 to January 6, 2017, we found that the $PM_{2.5}$ mass concentration increases
correspondingly with the lower RH.
In addition to the near-surface weak winds, persistent aerosol pollution is a result of temperature
inversions caused by the southwest warm advection (Fig. 10 (b, c), brown boxes). The ground of the



Qinghai-Tibet Plateau is a heat source throughout the year (Ye and Gao, 1979); thus, it heats the near-surface ambient air (Fig. 11). When the relatively warm air moves east across the Tibet Plateau under the southwesterly winds, it forms an inversion above the basin (Fig. 10 (c), brown boxes), which caps the convective layer and then induces the accumulation of aerosols and water vapor.

Effective pollution clearing rarely occurs in Chengdu because the Sichuan Basin is less affected by the cold, clean northerly winds as a result of the surrounding high northern mountains. However, as soon as aerosol pollutants and water vapor are cleared, aerosol pollution will form again due to more longwave radiation lost from the ground. For example, during the period of December 4-7, 2016, the fog/cloud dissipated, and the $PM_{2.5}$ mass concentration dropped to a low value on the 5th (Fig. 12 (a, d)). Due to the absence of cloud/fog blocking, more longwave radiation from the ground was emitted into space on the 6th night, and the surface net radiant exposure decreased from -0.58 on the 5th to -1.45 on the 6th (2.5 times) (Fig. 12 (e)). The significant reduction in the surface radiation cooled the near-surface atmospheric temperature, which formed an inversion layer of approximately 50~100 m (Fig. 12 (c)). Capped by the inversion layer, the $PM_{2.5}$ mass concentration doubled after the night of the 6th to form another aerosol pollution event ((Fig. 12 (a)).

Pollution removal in Chengdu mainly relies on northeasterly winds to blow pollution away. The winds also carry water vapor to add humidity to the atmosphere above Chengdu, which converts pollutants into fog/cloud drops or produces precipitation that removes pollutants through wet removal (Fig. 10 (d), blue boxes).

**3.6 The two-way feedback mechanism exists on the windy Northeast China Plain, where mid-lower warm, humid southwesterly winds transport aerosol pollutants from polluted southwestern regions, and strong, clean northwesterly winds blow pollutants away.**

The Northeast China Plain lies north of the Liaodong Gulf, west of the Changbai Mountains, east of the Greater Khingan, and south of the Lesser Khingan (Fig. 1). Due to the low mountains to the northwest, the Northeast China Plain is susceptible to cold, dry northerly air from Siberia in winter. As the largest city in Northeast China in terms of its urban population, Shenyang is located in the southwestern Northeast China Plain (Fig. 1), where the warm, humid southwesterly flows are transported from Bohai Bay.

From December 1, 2016 to January 10, 2017, six aerosol pollution episodes appeared in Shenyang



(Fig. 13 (a), blue boxes), four of which persisted for more than 3 days with peak $PM_{2.5}$ mass
concentrations greater than 200 μg m$^{-3}$ (Fig. 13 (a)). During these HPEs, we observed surface radiation
reductions, near-surface inversions, low-level RH enhancement, and increases in the $PM_{2.5}$ mass
concentration (Fig. 13 (a, c, d), red and white boxes) under slight or calm winds (Fig. 13 (b), red boxes);
these conditions indicate the occurrence of the two-way feedback mechanism in Shenyang.

Compared with those in Xi'an, Nanjing, Wuhan, Qingyuan, and Chengdu, the speeds of the

southeasterly or northwesterly winds are strikingly higher in Shenyang. Relatively strong mid-lower
southeasterly winds originate from Bohai Bay, and these winds transport warm, humid air that heats and
adds humidity to the mid-upper layer above Shenyang (Fig. 13 (c, d)). This air also transports aerosol
pollutants to Shenyang because it carries pollutants from populated and polluted southwestern industrial
regions, including Anshan. Lower strong northwesterly winds carry dry, cold air from Siberia to remove
pollutants in Shenyang (Fig. 13 (a, b)).
**3.7  Quantifying the two-way feedback mechanism and comparing its magnitude in various haze**
**regions of China.**

As previously mentioned, the weak two-way feedback mechanism in the Sichuan Basin is weakened

by the cloudy mid-upper layers, which compete with the near-surface aerosols for solar radiation.
However, the mechanism occurred in the Guanzhong Plain, the middle and lower reaches of the Yangtze
River, the Pearl River Delta region, and the Northeast China Plain. To quantify the magnitude of the two-
way feedbacks in these haze regions of China, we obtained the air temperature difference between the
radiosonde observations affected by this two-way feedback and the ERA-interim reanalysis data without
feedback in the regional center cities, including Beijing, Xi'an, Shenyang, Wuhan, Nanjing, and
Qingyuan (to replace Guangzhou). A previous study established a threshold value for the $PM_{2.5}$ mass
concentration (100 μg m$^{-3}$) that effectively activates the two-way feedback in HPEs; additionally, a lower
threshold value (71 μg m$^{-3}$) has been identified for lighter HPEs (Zhong et al, 2018, Tellus B, accepted).
Therefore, based on the diurnal mean $PM_{2.5}$ mass concentration (from 08:00 to 17:00 BJT), the
temperature difference is further classified by the criterion of 100 μg m$^{-3}$ in the more polluted North
China Plain, Guanzhong Plain, and Northeast China Plain and by the criterion of 71 μg m$^{-3}$ in the less
polluted Two Lakes Plain, Yangtze River Delta, and Pearl River Delta.

By comparing the air temperature difference below and above these thresholds in the six cities, we



found that the lower temperature profile was strikingly modified by the two-way feedback mechanism
(Fig. 14). On the North China Plain, the Guanzhong Plain, and the Northeast China Plain, the lower
temperature bias between the sounding observations and the ERA-interim data was close to zero below
the threshold of 100 μg m$^{-3}$ but immediately became negative above the threshold (Fig. 14 (a, b, c)). In
the Two Lakes Plain, the Yangtze River Delta, and the Pearl River Delta, we observed a similar reduction
in the temperature difference below and above the threshold of 71 μg m$^{-3}$ (Fig. 14 (b, c, d)). Overall, the
magnitude of the two-way feedback mechanism was larger in the North China Plain, the Guanzhong
Plain, and the Northeast China Plain than in the Two Lakes Plain, the Yangtze River Delta, and the Pearl
River Delta.

For each representative site, the low-level cooling bias was more striking near the ground surface;

additionally, as the $PM_{2.5}$ mass concentration increased, the low-level cooling bias became more
significant (Fig. 14). In Beijing, the negative temperature difference reached more than 2℃ with $PM_{2.5}$
values in the range of 200 ~ 300 μg m$^{-3}$ compared to approximately 1℃ in the range of 100 ~ 200 μg m
$^{-3}$. In Xi'an, the temperature difference decreased from approximately -1.5℃ in the range of 100 ~ 200
μg m$^{-3}$ to 2.5℃ in the range of 200 ~ 300 μg m$^{-3}$. In Shenyang, the cooling bias of approximately 0.6℃
occurred with the increase in $PM_{2.5}$ from 100 ~ 200 μg m$^{-3}$ to 200 ~ 300 μg m$^{-3}$. Under the most polluted
conditions, the near-ground cooling bias was greater than -4℃, approximately -4℃, and approximately
-1℃ in Beijing, Xi'an, and Shenyang, respectively, which was substantially affected by the two-way
feedback.

To quantify the feedback of the worsened meteorological conditions on the increasing $PM_{2.5}$ in the

CSs, a PLAM index was used, which mainly reflects the stability of the air mass and the condensation
rate of water vapor on aerosol particles. The squared correlation coefficients between the hourly PLAM
and $PM_{2.5}$ mass concentration in the typical $PM_{2.5}$ increase processes during the CSs were 0.71, 0.7, 0.72,
0.68, 0.64, and 0.63 in Beijing, Xi'an, Shenyang, Wuhan, Nanjing, and Qingyuan, respectively (Fig. 15
(a, b, c, d, e, f)); these values exceeded the 0.05 significance level, which suggested that such a
meteorological feedback on $PM_{2.5}$ explained 60~70% of the increase in the $PM_{2.5}$ during the CSs.
**4 Conclusions:**

Here, we used $PM_{2.5}$ observations, surface radiation data, radiosonde observations, meteorological





index-PLAM, and ERA-interim reanalysis data to investigate the formation, accumulation, and
dispersion of aerosol pollution during persistent heavy aerosol pollution episodes over 3 days (HPEs),
particularly focusing on the two-way feedback mechanism between the unfavorable meteorological
conditions and the cumulative $PM_{2.5}$ pollution in various haze regions in China, including the Guanzhong
Plain, the Yangtze River Delta region, the Two Lakes Basin, the Pearl River Delta, and the Sichuan Basin.
On the Guanzhong Plain, we observed a striking two-way feedback mechanism, including reduced
surface radiations, near-surface inversions, RH enhancement in the lower part of BL, and increases in
$PM_{2.5}$ mass concentrations under slight or calm winds in the CSs. For the representative sites of Xi'an,
the near-ground cooling bias caused by the two-way feedback was as high as approximately -4 ºC, which
was similar to that observed in Beijing. Bordered by the Qinling Mountains and the Loess Plateau, the
Guanzhong Plain experiences inter-regional pollution transport below the BL, e.g., pollution transport to
Xi'an from Yuncheng and Linfen under lower northwesterly winds in the TSs. Pollution clearing mainly
depends on the lower strong northeasterly winds to blow pollutants away and the mid-upper southerly
winds to transport water vapor to increase RH, which causes the $PM_{2.5}$ to enter the fog-cloud phase.
In the relative less polluted Yangtze River Delta region, the aerosol pollution formation is similar to
that in Beijing, including earlier TSs and later CSs. During the TSs, the Yangtze River Delta region is
affected by trans-regional pollution transport below and over the BL from the North China Plain, which
induces increases in the $PM_{2.5}$ in near surface or at the higher atmosphere in this region, which includes
Nanjing and Shanghai. Upper transported pollutants would move downward to further worsen the near-
ground aerosol pollution. During the CSs, we also observed the two-way feedback mechanism, but its
magnitude is lower than that in Beijing due to the less-polluted conditions. In this region, pollution
clearing relies on persistent stronger northerly winds bring pollutants out of this area, or strong
southeasterly winds, which transport clean, warm, humid air that blows pollutants away or increase
ambient RH to cause the $PM_{2.5}$ to enter the liquid fog-cloud phase. Similar to the Yangtze River Delta
region, the Two Lakes Basin also experienced trans-regional pollution transport from the North China
Plain under northerly winds below and sometimes over the BL during the TSs. During the CSs, the two-
way feedback is activated and the aerosol pollution worsens. In addition to the blowing effect of strong,
persistent northerly winds, pollution clearing also depends on the mid-upper southerly winds, particularly
the southwesterly winds, to transport water vapor, which enhances the RH and eliminates pollutants



through fog-cloud conversion and wet removal.
In the least polluted Pearl River Delta, no feedback mechanism was observed with $PM_{2.5}$ mass
concentrations below the threshold. However, when the $PM_{2.5}$ concentration exceeded the threshold, the
two-way feedback occurred in the CSs. The delta region was purified by lower clean, cold northeasterly
winds from the northern mountains and humidified by upper southerly winds from the South China Sea.
The Sichuan Basin is dominated by high RH and weak winds; thus, the two-way feedback
mechanism was weakened by thick mid-upper fog/clouds that compete with the near-surface aerosols for
solar radiation and consequently cool the whole atmosphere below. With the weak two-way feedback,
the $PM_{2.5}$ mass concentration increased under lower slight or calm winds and was capped by the upper
temperature inversions caused by the upper southwesterly winds from the Tibet Plateau. Pollution
clearing mainly relies on northeasterly winds to blow pollutants away, and these winds also add humid
air to the atmosphere, which converts aerosols into fog/cloud drops. Although pollutants and water vapor
are cleared, aerosol pollution will soon form again due to more longwave radiation lost from the ground,
which results in rare effective pollution clearing in the Sichuan Basin.
Compared with the above regions, the southerly and northerly winds are strikingly larger in the
Northeast China Plain. Strong mid-lower southeasterly winds originate from Bohai Bay, transport warm,
humid air that heats and adds humidity to the inland area and transport pollutants inter-regionally from
polluted southwestern industrial regions. Lower strong northwesterly winds carry dry, cold air from
Siberia to remove pollutants. At the representative site in Shenyang, a two-way feedback mechanism also
exists during the CSs with slight or calm winds.
The transport, accumulation and removal of pollution described above is visually illustrated in a
conceptual scheme (Fig. 16), which particularly highlights the effect of the two-way feedback mechanism
in the role of intensifying the HPEs. Due to the occurrence of a two-way feedback mechanism, effective
pollution control could further mitigate aerosol pollution, while persistent worsening aerosol pollution
could lead to an additional increase in $PM_{2.5}$. Given the inter-regional and trans-regional pollution
transport, the control of regional emissions among key haze regions in China, to reduce the pollutants
transport or to let them not reach the threshold enough to trigger two-way feedback mechanism, is
essential to substantially reduce persistent heavy aerosol pollution episodes. At the same time, these
results also show that, even in favorable weather conditions, aerosol pollutant emissions should not be





allowed to occur without restrictions; when aerosol pollution cumulates to a certain extent, it will
significantly worsen the BL meteorological conditions and "close" the "meteorological channels"
available for pollution dispersion.



**Acknowledgment:**

This research is supported by the National Key Project of MOST (2016YFC0203306) and the

Atmospheric Pollution Control of the Prime Minister Fund (DQGG0104).
**Author Contributions:**

X.Y.Z. and Y.Q.W. designed the research; X.Y.Z and J.T.Z carried out the analysis of observations.

J.Z.W provided PLAM data. X.J.S conducted a supplementary analysis. J.T.Z. wrote the first manuscript
and X.Y.Z. revised it. All authors contributed to the improvement of this manuscript and approved the
final version.
**Additional Information:**

Competing financial interests: The authors declare no competing financial interests.






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



**Figure captions:**

**Figure 1: The key haze regions with similar declines in visibility in eastern China.** (White dot: locations of radiosonde stations)

**Figure 2: National distribution of mean PM$_{2.5}$ mass concentration from December 1, 2016 to January 10, 2017.** (White dot: locations of radiosonde stations)

**Figure 3: Temporal variations in PM$_{2.5}$, surface radiation, and vertical distributions of meteorological factors from December 1, 2016 to January 10, 2017.** (a) PM$_{2.5}$ mass concentration (gray line: Beijing; light gray line: Yuncheng); (b) winds (vectors) and wind velocity (shadings; units: m s$^{-1}$); (c) temperature (shadings; units: °C); (d) RH (shadings; units: %); and (e) global radiant exposure. (Blue line: HPEs; light blue line: PEs; white or red box: CSs; and brown box: water vapor transport)

**Figure 4: Temporal variations in PM$_{2.5}$, surface radiation, and vertical distributions of meteorological factors from December 1, 2016 to January 10, 2017.** (a) PM$_{2.5}$ mass concentration (gray line: Nanjing; light gray line: Jinan); (b) winds (vectors) and wind velocity (shadings; units: m s$^{-1}$); (c) temperature (shadings; units:°C); (d) RH (shadings; units: %); and (e) global radiant exposure. (Blue line: HPEs; light blue line: PEs; white or red box: CSs; and blue dot: precipitation.)

**Figure 5: The distribution of the concentration differences in PM$_{2.5}$ mass between the start time and the end time (the end is subtracted from the start) of the TSs in Figure 4.** (a) TS$_1$ in HPE$_1$; (b) TS$_2$ in HPE$_1$; (c) TS in HPE$_2$; and (d) TS in HPE$_3$.

**Figure 6: Temporal variations in PM$_{2.5}$ and vertical distributions of meteorological factor from December 1, 2017 to December 9, 2017.** (a) PM$_{2.5}$ mass concentration (gray line: Nanjing; light gray line: Xingtai); (b) winds (vectors) and wind velocity (shadings; units: m s$^{-1}$); (c) temperature (shadings; units: °C); and (d) RH (shadings; units: %). (orange box: TSs; purple box: clean periods)

**Figure 7: Temporal variations in PM$_{2.5}$, surface radiation, and vertical distributions of meteorological factors from December 1, 2016 to January 10, 2017.** (a) PM$_{2.5}$ mass concentration (gray line: Shanghai; light gray line: Nanjing); (b) winds (vectors) and wind velocity (shadings; units: m s$^{-1}$); (c) temperature (shadings; units: °C); (d) RH (shadings; units: %); and (e) direct radiant exposure (of the vertical surface to the direction of solar radiation) and global radiant exposure. (White or red box: CSs; orange box: TSs; and blue dot: precipitation.)

**Figure 8: Temporal variations in PM$_{2.5}$, surface radiation, and vertical distributions of meteorological factors from December 1, 2016 to January 10, 2017.** (a) PM$_{2.5}$ mass concentration (gray line: Wuhan); (b) winds (vectors) and wind velocity (shadings; units: m s$^{-1}$); (c) temperature (shadings; units: °C); (d) RH (shadings; units: %); and (e) direct radiant exposure (of the vertical surface to the direction of solar radiation) and global radiant exposure. (Blue line: HPEs; light blue line: PEs; white or red box: CSs; orange box: TSs; brown box: water vapor transport; and blue dot: precipitation.)

**Figure 9: Temporal variations in PM$_{2.5}$, surface radiation, and vertical distributions of meteorological factors from December 1, 2016 to January 10, 2017.** (a) PM$_{2.5}$ mass concentration (gray line: Qingyuan; light gray line: Guangzhou); (b) winds (vectors) and wind velocity (shadings; units: m s$^{-1}$); (c) temperature (shadings;



units: ℃); (d) RH (shadings; units: %); and (e) direct radiant exposure (of the vertical surface to the direction of
solar radiation) and global radiant exposure. (Blue line: HPEs; white or red box: CSs; purple box: clearing; brown
box: water vapor transport; and blue dot: precipitation.)

**Figure 10: Temporal variations in PM$_{2.5}$, surface radiation and vertical distributions of meteorological**
**factors from December 1, 2016 to January 10, 2017.** (a) PM$_{2.5}$ mass concentration (gray line: Chengdu); (b)
winds (vectors) and wind velocity (shadings; units: m s$^{-1}$); (c) temperature (shadings; units: ℃); (d) RH (shadings;
units: %); and. (Blue line: HPEs; white or red box: CSs; brown box: warm air flow or inversions; and blue dot:
precipitation.)

**Figure 11: Vertical section of mean air temperature in December 2016 at 30.67°N.**

**Figure 12: Temporal variations in PM$_{2.5}$, surface radiation and vertical distributions of meteorological**
**factors from December 4 to 7, 2017.** (a) PM$_{2.5}$ mass concentration (gray line: Chengdu); (b) winds (vectors) and
wind velocity (shadings; units: m s$^{-1}$); (c) temperature (shadings; units: ℃); (d) RH (shadings; units: %); and (e)
global radiant exposure.

**Figure 13: Temporal variations in PM$_{2.5}$, surface radiation and vertical distributions of meteorological**
**factors from December 1, 2016 to January 10, 2017.** (a) PM$_{2.5}$ mass concentration (gray line: Shenyang); (b)
winds (vectors) and wind velocity (shadings; units: m s$^{-1}$); (c) temperature (shadings; units: ℃); (d) RH (shadings;
units: %); and (e) global radiant exposure. (Blue line: HPEs; light blue line: PEs; white or red box: CSs; and blue
dot: precipitation.)

**Figure 14: Vertical temperature difference between sounding observations and ERA-interim reanalysis data**
**under different concentration bins of PM$_{2.5}$ mass (µg m$^{-3}$).** (a) Beijing; (b) Xi'an; (c) Shenyang; (d) Wuhan; (e)
Nanjing; and (f) Qingyuan.

**Figure 15: Correlation between PLAM and PM$_{2.5}$ during the typical rising processes of PM$_{2.5}$ from**
**December 1, 2016 to January 10, 2017.**

**Figure 16: A concept scheme of pollution removal (a), transport (b), and accumulation (c), particularly the**
**two-way feedback mechanism between the unfavorable meteorological conditions and the cumulative**
**aerosol pollution (c).**



**Figure 1**

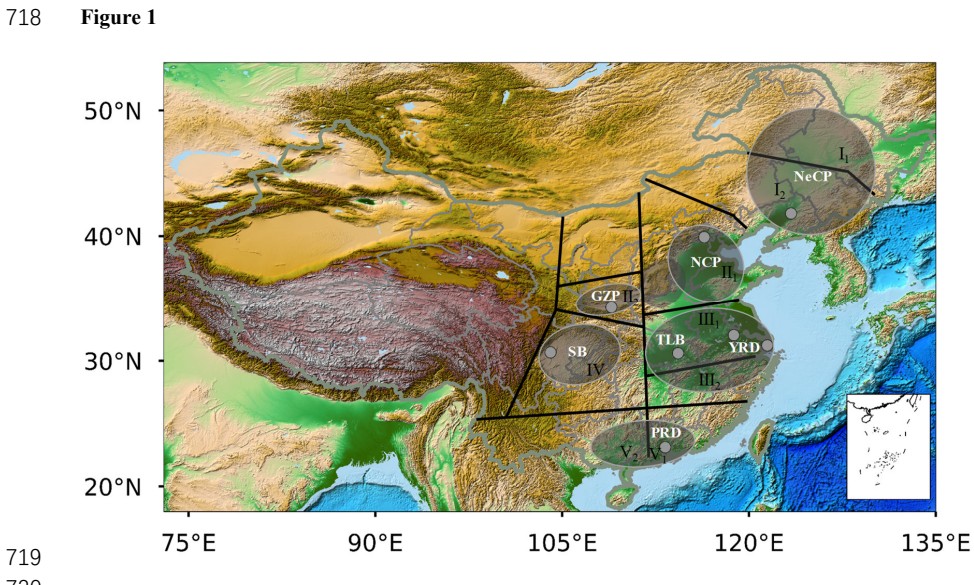






**Figure 2**

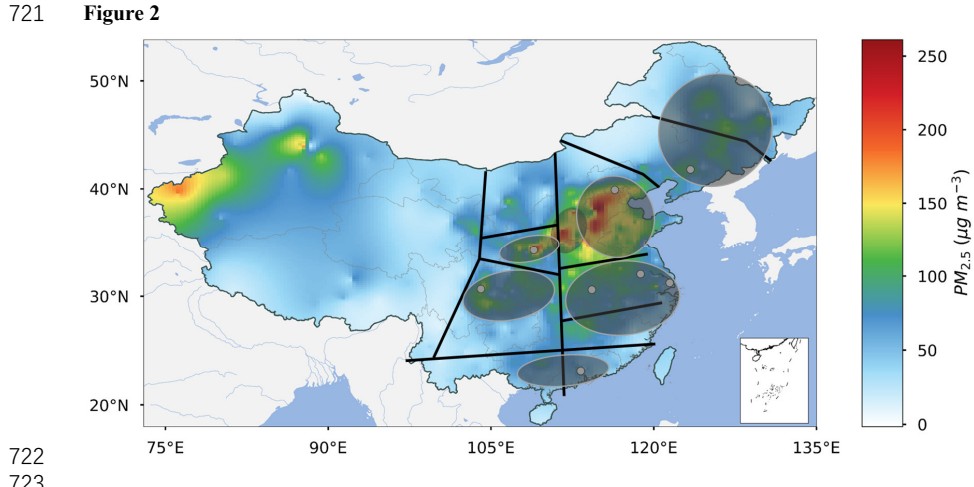




**Figure 3**

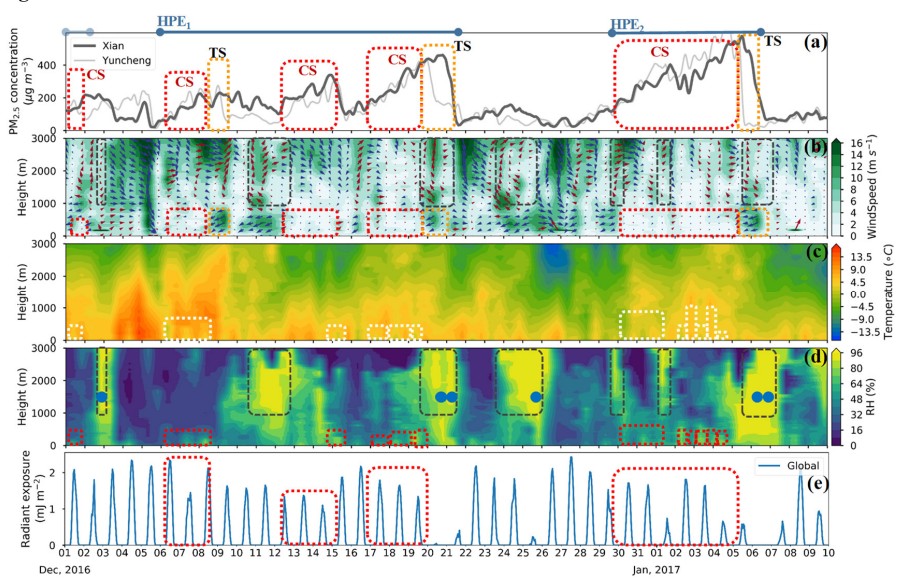




**Figure 4**

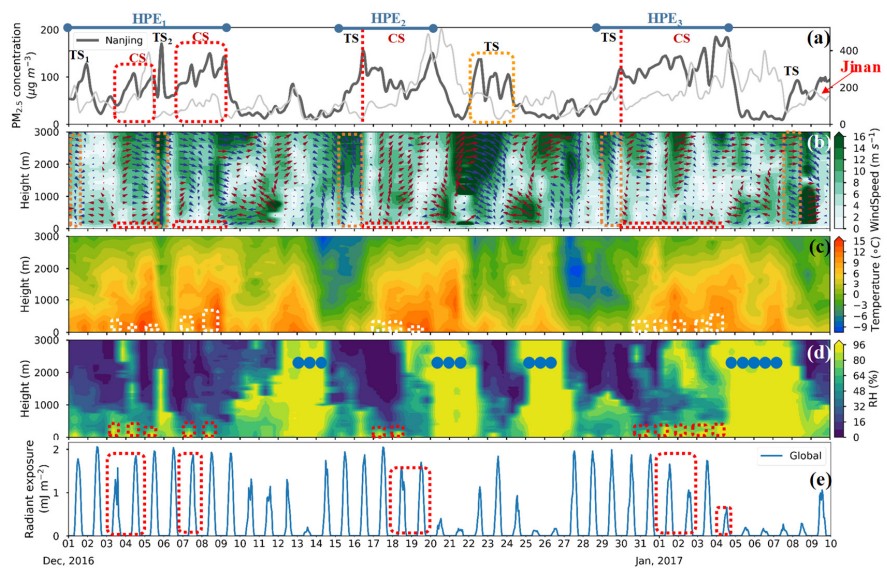






**Figure 5**

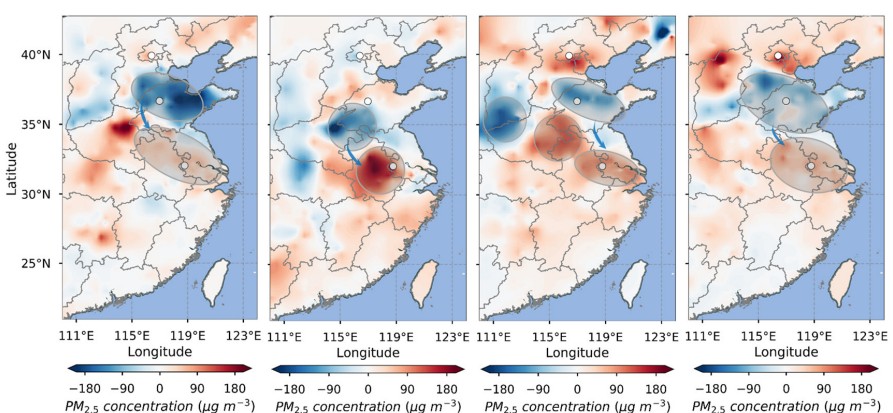





**Figure 6**

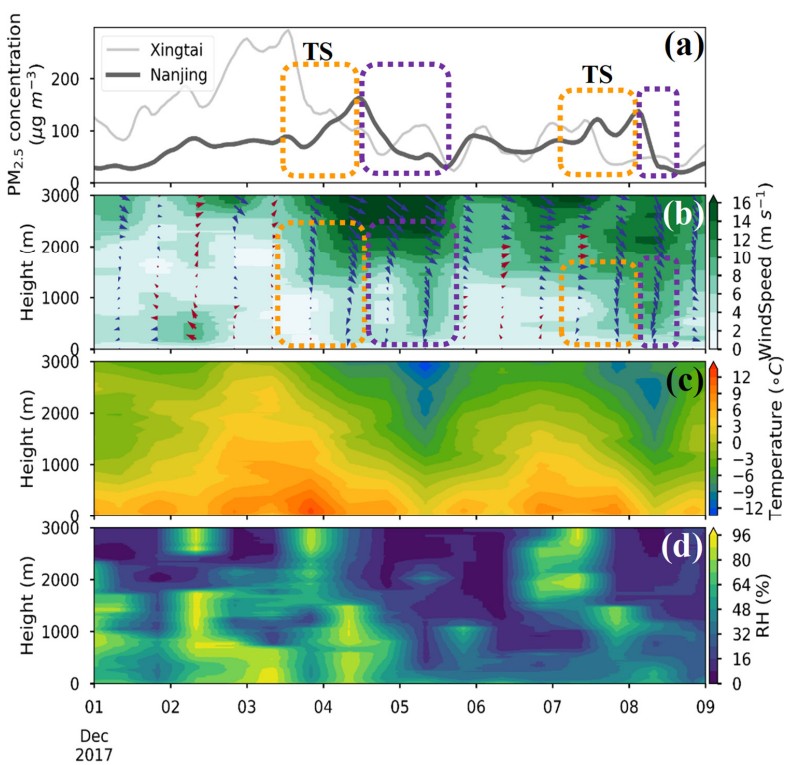



**Figure 7**

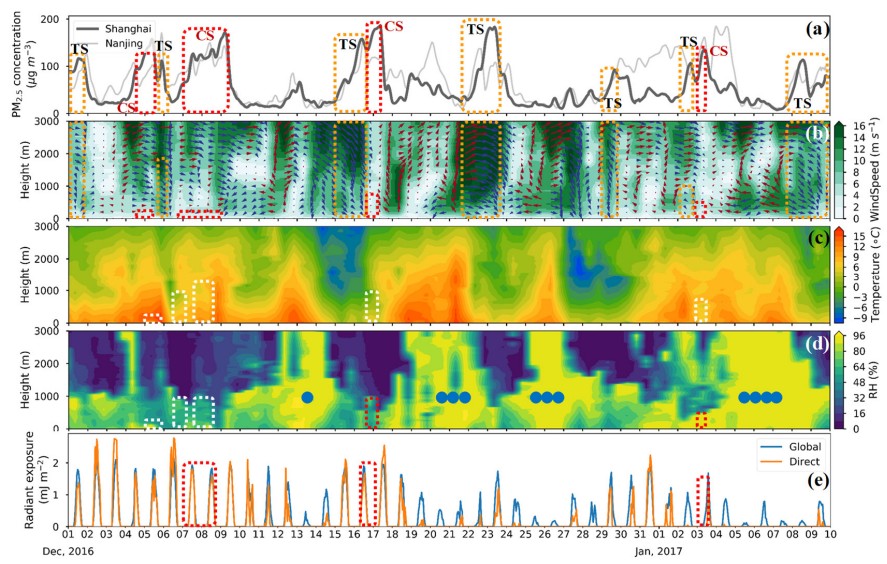






**Figure 8**

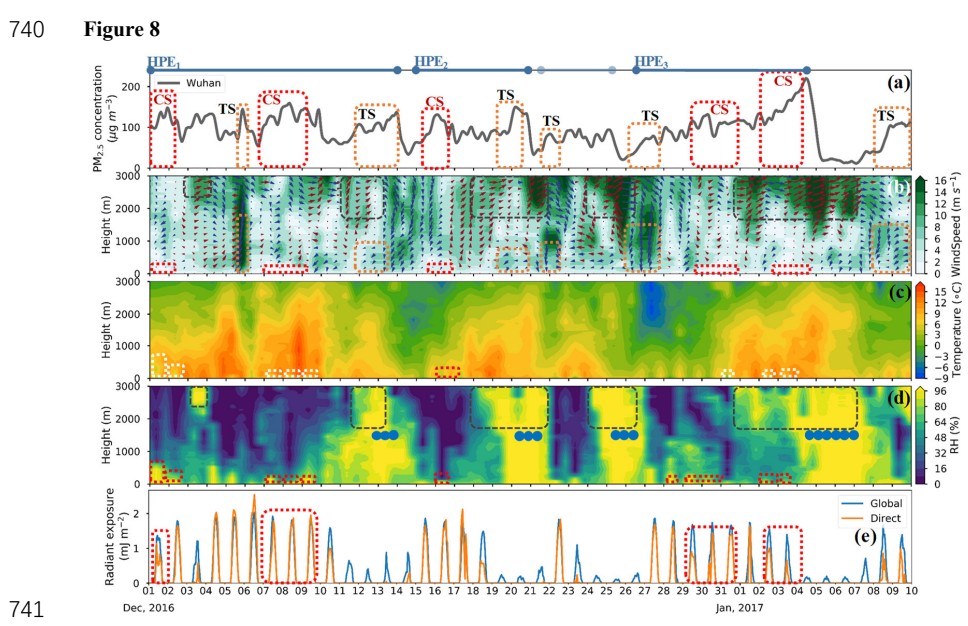




**Figure 9**

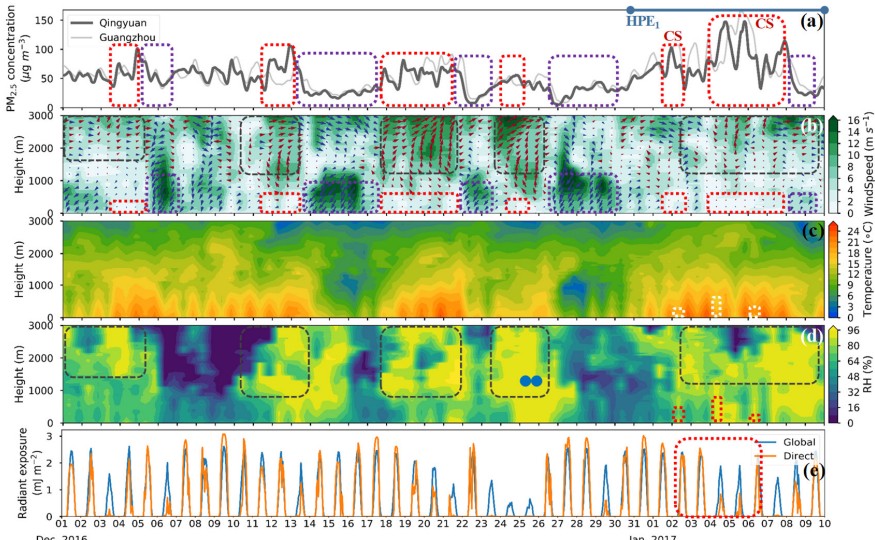





**Figure 10**

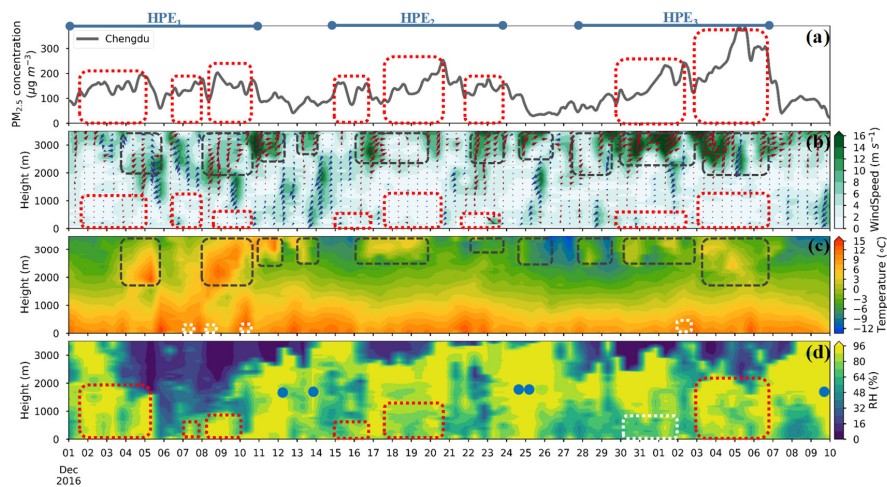






**Figure 11**

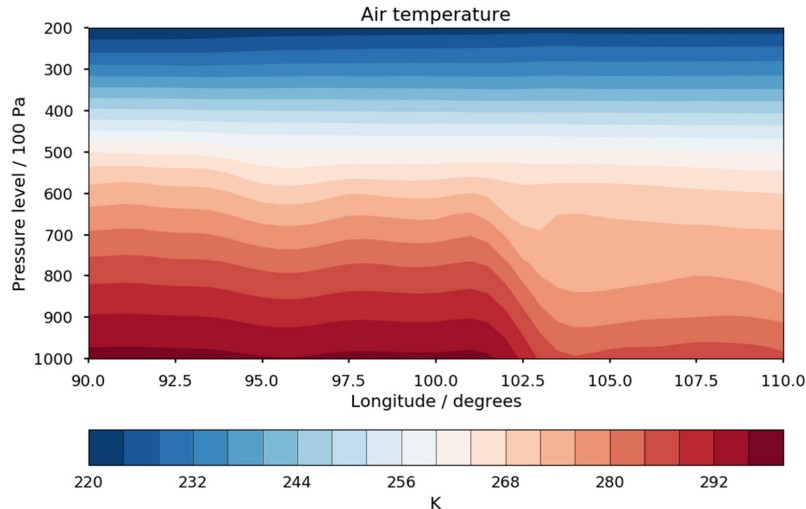





**Figure 12**

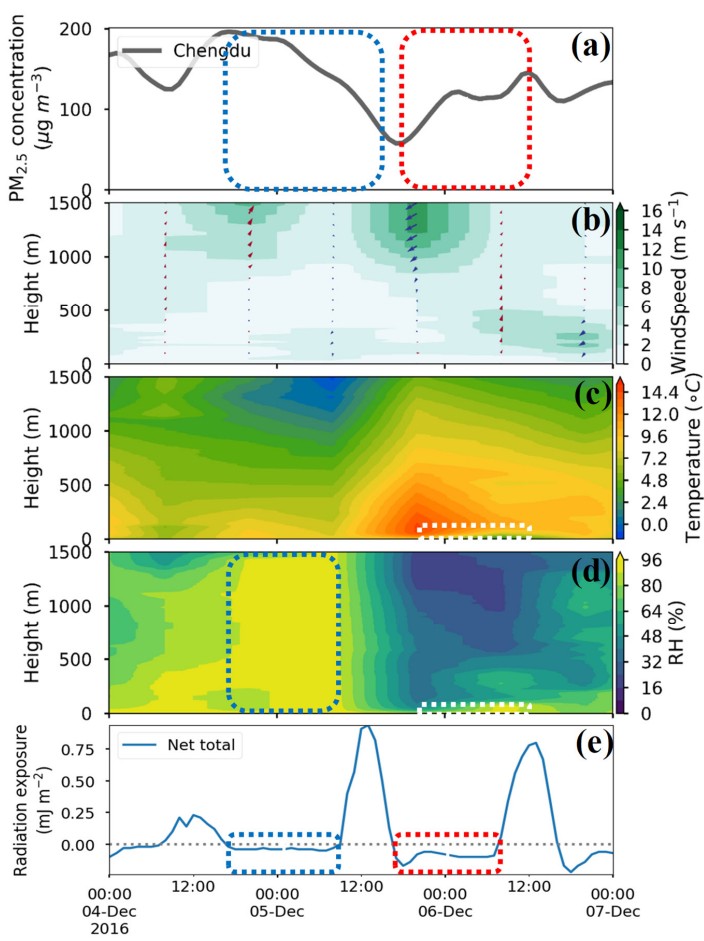





**Figure 13**

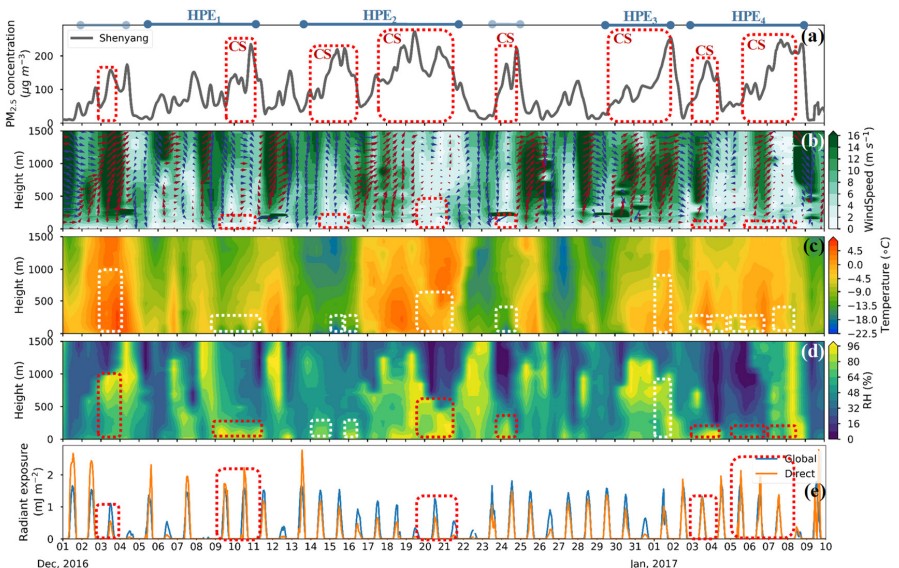






**Figure 14**

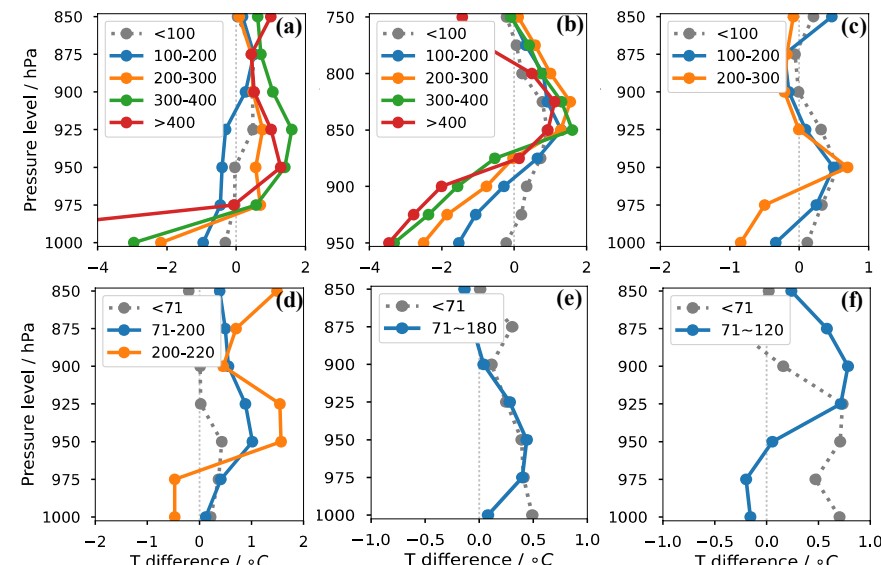





**Figure 15**

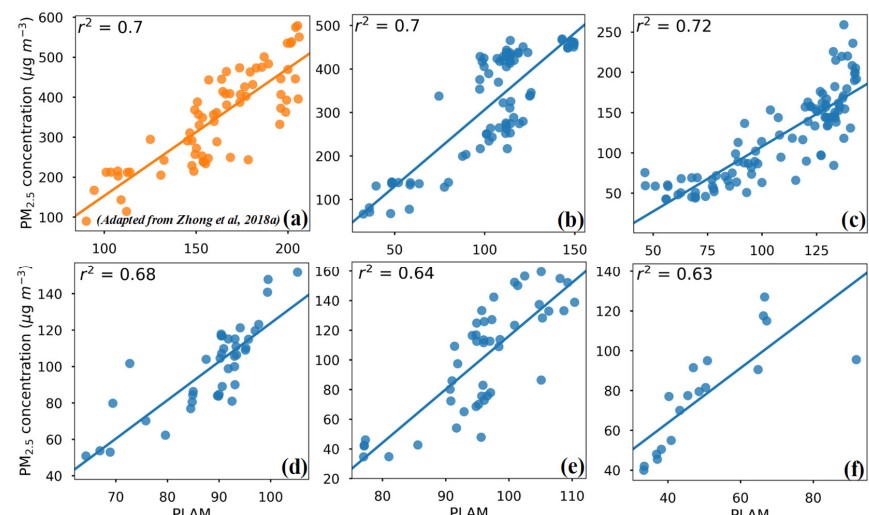





**Figure 16**

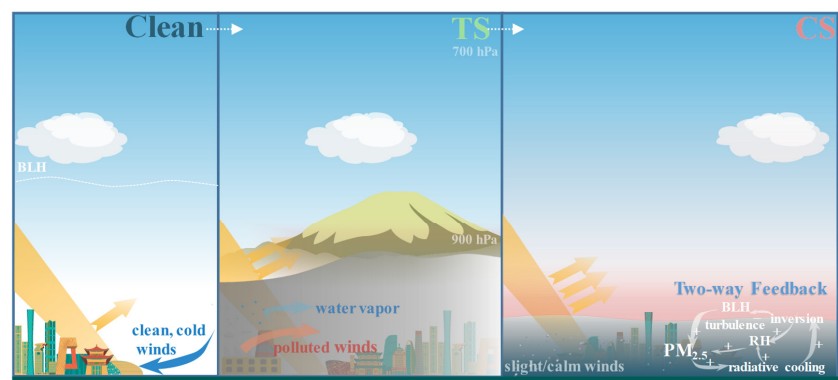
