# Peer review of "Two-way feedback mechanism between unfavorable"

_Atmospheric Chemistry and Physics, 2018_

## Referee Comment (RC1) · Anonymous Referee #1 · 1 Dec 2018

This work tried to understand the role of two-way feedback between unfavorable meteorological condition and air pollution based on in-situ measurements on both air quality as well as reanalysis data. The authors also compared this feedback mechanism in different highly-polluted regions in China. The strength of this work is comprehensive and high-resolution observational data in multiple typical regions. However, more in-depth analysis and quantitative discussion ought to be provided while interpreting the disparities of two-way feedback in different regions and in temporal stage, as well as its relationship with other factors. Here are some issues that need to be addressed.

[Figure]

Major comments:

Most parts of Results and Discussion, i.e. Section 3.1-3.6, describe measurements on temporal variations in PM2.5, surface radiation, and vertical distributions of meteorological conditions like air temperature and humidity. Indeed, the literal description and full-information figures give a detailed picture of air pollution and meteorological conditions for each individual regions during these 40-day period. However, I personally think that further in-depth analysis and discussions need to be performed for the purpose of better understanding this two-way feedback in various regions and also in different stages during air pollution. For instance, the authors clearly identified different pollution stage in different regions in the manuscript, including cumulative stages, transport stage as well as clean stage. What are the quantitative differences of the two-way feedback mechanism between transport stage and cumulative stage? How the different synoptic conditions (wind, radiation, cloud and humidity) during different stages influence the pollution vertical profile and then the feedback processes?

Another, in Section 3.7 where statistical analysis is made to comprehend the disparities of the two-way feedback under different pollution conditions, more relevant parameters and quantitative results is suggested. Since that vertical differences in temperature stratification between observations and reanalysis data are presented in Fig. 14, the aerosol profile, which has been identified to play an important role in radiative effect of aerosol (Wilcox et al.,2016; Wang et al, 2018), is better to be discussed. If possible, this work will be further improved by including some measurements on aerosol extinction profiles.

Given that this work mainly probed into "two-way feedback mechanism" in the whole manuscript, it is better to clearly define this term before the result part. What are exactly the included chemical and physical processes? Does it mean aerosol self-induced pollution deterioration, or it has taken synoptic condition into account already? The authors indicated that meteorological feedback explained 60-70% pollution increase in Line 458-459. Does it mean "two-way feedback mechanism" that has already included unfavorable meteorological condition caused by synoptic weather condition? If so, "meteorological feedback" is a little misleading.

This work focused on two-way feedback mechanism on the time period between Dec. 1 2016 to Jan. 10 2017. Please give the reasons for choosing this 40-day data since that both the air quality monitoring data and reanalysis data cover much longer period than that being used in this work.

Minor corrections:

Please define the abbreviation in the main text for the first time and do not repeatedly explain it in the following part. And abbreviations should not be included without explanations in the Abstract. It needs to be thoroughly checked. (Line 27, Line 214, Line 244...)

The title for each subsection is a little bit long. Please simplify it in the revision.

Section 2.1: The reference or data URL ought to be provided here.

Line 101-102: This sentence, which describes data utilization in this work, is better to be placed in Section 2.

Line 203-206: The authors proposed enhanced hygroscopic growth of aerosol due to increase in RH. This point could be further confirmed by the ratios of Lidar-observed aerosol extinction and PM2.5 concentration provided by air quality monitoring net work. Thus, more information is suggested to be added here.

Line 209-201: Regional transport of air pollution from Yuncheng to Xi'an is claimed here. Simple backward trajectories is recommended to be included here to clearly show the transport pathway.

References:

Wilcox, E. M. , Thomas, R. M. , Praveen, P. S. , Pistone, K. , Bender, F. A. M. , & Ramanathan, V. . (2016). Black carbon solar absorption suppresses turbulence in the

atmospheric boundary layer. Proc Natl Acad Sci U S A, 113(42), 11794-11799.

Wang, Z., Huang, X., & Ding, A. (2018). Dome effect of black carbon and its key influencing factors: a one-dimensional modelling study. Atmospheric Chemistry & Physics, 18(4), 1-29.

---

## Referee Comment (RC2) · Anonymous Referee #2 · 2 Jan 2019

This paper uses very comprehensive measurements, including PM2.5 mass concentrations, radiosonde observations, etc., to understand the two-way feedback mechanism between unfavorable meteorological conditions and cumulative aerosol pollution in several heavily and less heavily polluted regions in China. Previous studies have been focused on the North China region, while this study can provide a more general picture with extended study regions. My major concern for the paper is that the presentations of the results are descriptive, and I would suggest the authors provide more in-detailed calculations and present in a more quantitative way.

[Figure]

Specific comments are listed below: 1 Near ground temperature bias are shown at different PM2.5 concentrations, however, it is not clear to what extend is contributed by aerosol effects. It is likely that those temperature structures co-occur with different stagnant conditions and thus are different at different PM2.5 concentrations. It would be better to provide more quantitative results here.

2 Line 41-43: It is not surprising that FWRP and YRD regions are largely influenced by inter-regional and trans-regional transport. It would be more valuable to know in these regions, how important are the two-way feedback in cumulative periods, compared to the abrupt injection of transport.

---

## Author Comment (AC1) · 13 Jan 2019

Authors reply to reviewer's comments:

Dear Anonymous Referee,

Thanks for your careful review of the manuscript. We read the reviewer's comments carefully, and have responded and taken all of reviewer's comments into consideration and revised the manuscript accordingly. My detailed responses, including a point-by-point response to the review and a list of all relevant changes, are as follows:

[Figure]

"Reviewer #1: This work tried to understand the role of two-way feedback between un-favorable meteorological condition and air pollution based on in-situ measurements on both air quality as well as reanalysis data. The authors also compared this feedback mechanism in different highly-polluted regions in China. The strength of this work is comprehensive and high-resolution observational data in multiple typical regions. However, more in-depth analysis and quantitative discussion ought to be provided while interpreting the disparities of two-way feedback in different regions and in temporal stage, as well as its relationship with other factors. Here are some issues that need to be addressed. "1) Most parts of Results and Discussion, i.e. Section 3.1-3.6, describe measurements on temporal variations in PM2.5, surface radiation, and vertical distributions of meteorological conditions like air temperature and humidity. Indeed, the literal description and full-information figures give a detailed picture of air pollution and meteorological conditions for each individual regions during these 40-day period. However, I personally think that further in-depth analysis and discussions need to be performed for the purpose of better understanding this two-way feedback in various regions and also in different stages during air pollution. For instance, the authors clearly identified different pollution stage in different regions in the manuscript, including cumulative stages, transport stage as well as clean stage. What are the quantitative differences of the two-way feedback mechanism between transport stage and cumulative stage? How the different synoptic conditions (wind, radiation, cloud and humidity) during different stages influence the pollution vertical profile and then the feedback processes?"

Response: Thanks for the comments. We have added Fig. 5 and Fig.16, together with corresponding discussions to quantitatively estimate the magnitude of the two-way feedback from TSs to CSs in different regions (L242-L260; L514-524). To better show the aerosol-induced temperature reduction, we mainly focused on polluted Beijing, Xi'an, and Shenyang, which have more striking meteorological modification. In addition, the two-way feedback has been defined in detail, including the different impact of wind, radiation, and humidity during different stages (L66-92). In addition, Fig. 1 has been added in the manuscript from the supplement to better illustrate above processes. During the time period in Fig.1, the impact of clouds on solar radiation has been discussed in the previous investigation as follows (Zhong et al., 2018b): The factors that cause such radiation reduction from the clean stages to the TSs to the CSs include high-layer moisture (> 1000 m), low-layer moisture (< 1000 m) and radiative cooling of aerosols (Ding et al., 2016; Wang et al., 2015). During the four HPEs in Fig. 1, the high-layer moisture is relatively low with RH almost less than 30% from the clean stages to the TSs to the CSs, which suggests the contribution of high-layer moisture (including water vapor and liquid water) is limited to noted radiation reduction. Distinctive from high-layer moisture, the low-layer moisture (including water vapor and liquid water) strikingly increases from the clean stages to the TSs due to wind shifted from northerly to southerly, because northwesterly winds, which originate from less populated northern mountainous areas, carry dry and cold air masses while warm and humid southerly winds transport more water vapor to Beijing (Guo et al., 2014; Jia et al., 2008; Liu et al., 2013; Zhong et al., 2017a). During the CSs with weak winds, appreciable near-surface moisture accumulation appears with RH over 80% (Fig. 1 c, d). Such enhanced moisture from the clean stages to the TSs to the CSs would reduce direct radiation through accelerating liquid-phase and heterogeneous reactions (Cheng et al., 2016; Tie et al., 2017; Wang et al., 2016) to produce more secondary aerosols and enhancing aerosol hygroscopic growth to increase aerosol particle size and mass (Kuang et al., 2016), which would back-scatter more solar radiation to space. Aerosol water serves as a medium that enables aqueous-phase reactions.

"2) Another, in Section 3.7 where statistical analysis is made to comprehend the disparities of the two-way feedback under different pollution conditions, more relevant parameters and quantitative results is suggested. Since that vertical differences in temperature stratification between observations and reanalysis data are presented in Fig. 14, the aerosol profile, which has been identified to play an important role in radiative effect of aerosol (Wilcox et al.,2016; Wang et al, 2018), is better to be discussed. If possible, this work will be further improved by including some measurements on aerosol extinction profiles."

Response: We added quotations and discussions about the radiative cooling effects of vertically distributed scattering and absorbing aerosols (L537-L543), which affect the temperature profile, particular the near-ground temperature stratification. The extinction profiles of aerosols were observed during Winter 2016/17 in Beijing, which indicates that aerosols were accumulated below $\sim$ 350 m during CSs and reach its max value near the ground(Zhong et al., 2018a). The aerosol profile affected the temperature stratification and resulted in the greatest temperature bias on the ground. We also wanted to show corresponding aerosol extinction profiles in other regions, which would definitely improve understanding on aerosol's radiative effects. However, unfortunately, aerosol extinction observations were insufficient in other regions except Beijing. Currently, we only have vertical PM2.5 mass concentrations in Nanjing, which were observed by an unmanned aerial vehicle from 3 December 2017 to 4 December 2017, so we added the vertical observations and corresponding discussions (L128-129; L322-325, Fig. 8).

"3) Given that this work mainly probed into "two-way feedback mechanism" in the whole manuscript, it is better to clearly define this term before the result part. What are exactly the included chemical and physical processes? Does it mean aerosol self-induced pollution deterioration, or it has taken synoptic condition into account already? The authors indicated that meteorological feedback explained 60-70% pollution increase in Line 458-459. Does it mean "two-way feedback mechanism" that has already included unfavorable meteorological condition caused by synoptic weather condition? If so, "meteorological feedback" is a little misleading."

Response: As suggested, the two-way feedback mechanism has been defined in the manuscript (L66-93). The details of the two-way feedback mechanism between unfavorable meteorological conditions and cumulative aerosols in Beijing are as follows: 1) When upper zonal large-scale circulations unfavorable for pollution dispersion occur, the boundary layer (BL) height reduces from $\sim$ 1500 m in clean stages to 700-800 m; under the BL, the winds shift from northerly to southerly, which transport pollutants

from the south of Beijing (transport stages (TSs) in Fig. 1). Above unfavorable meteorological conditions cause aerosol pollution formation. 2). When the vertical aerosols are accumulated to a certain degree, the dominant scattering aerosols will substantially back-scatter solar radiation, causing a reduction in the amount of solar radiation that reaches the surface, which causes a near-ground cooling effect through atmospheric circulation and vertical mixing (i.e., the cumulative sum of hourly radiant exposure reduced by 89% and 56%, respectively, from clean stages to cumulative stages (CSs) (Fig.1)) (Zhong et al., 2018b; Zhong et al., 2017b). With less solar radiation, near-ground temperature subsequently decreases. 3).Under slight or calm winds, the temperature reduction induces or reinforces an inversion that further weakens turbulence diffusion and results in a lower BL height, which further worsens aerosol pollution (during CSs in Fig.1). 5). This condition also decreases the near-ground saturation vapor pressure and suppresses water vapor diffusion to increase the relative humidity (RH), which will further enhances aerosol hygroscopic growth and accelerates liquid-phase and heterogeneous reactions to worsen aerosol pollution (Ervens et al., 2011; Kuang et al., 2016; Pilinis et al., 1989; Zhong et al., 2018a; Zhong et al., 2018b). This feedback effect of further worsened meteorological conditions aggravates PM2.5 pollution (during CSs in Fig.1) (Zhong et al., 2017b). Cumulative aerosols further worsen meteorological conditions, including RH increase, to further enhance aerosol hygroscopic growth and accelerates liquid-phase and heterogeneous reactions. These additional physical and chemical processes were included in the two-way feedback mechanism. HPEs generally included the TSs, whose aerosol pollution formation is primarily caused by pollutants transported from polluted regions, and the CSs, in which the PM2.5 increase is dominated by stable atmospheric stratification characteristic near-ground anomalous inversion, moisture accumulation and reduced BL height under slight or calm winds. During the CSs, the temperature inversion was found to be caused or reinforced mainly by accumulated aerosols. Other factors, including topography, advection and long-wave radiation are likely conducive to weak/normal inversion, but not dominant with respect to anomalous inversion. Therefore, the impact of synoptic conditions on aerosol pollution formation were mainly taken into account during the TSs, aerosol self-induced pollution deterioration play a more important role in the CSs. Compared with clean stages and TSs, the contributions of cumulative aerosols to further worsened BL meteorological conditions, including reinforced inversion, moisture enhancement and BL height are more striking than synoptic weather condition during the CSs. Therefore, during the CSs, the PLAM index dominantly include aerosol-induced meteorological changes.

"4) This work focused on two-way feedback mechanism on the time period between Dec.1 2016 to Jan. 10 2017. Please give the reasons for choosing this 40-day data since that both the air quality monitoring data and reanalysis data cover much longer period than that being used in this work'"

Response: This time period is consistent with the investigated HPEs in Beijing. Since the two-way feedback in other regions is based on previous work about HPEs in Beijing, using the same time period is convenient to compare aerosol pollution in Beijing and other regions. The time period from 1 December 2016 to 10 January 2017 was selected due to the following reasons. Based on the urban PM2.5 monthly mean mass concentration in Beijing in Winter 2016/2017, December 2016, which had the highest mass concentration was selected to represent the heavyPM2.5 pollution conditions in winter. As shown in Fig.1, the last HPE in December 2016 ended in 9 January 2017. Therefore, the time period from 1 January 2017 to 10 January 2017 was also selected.

"5) Please define the abbreviation in the main text for the first time and do not repeatedly explain it in the following part. And abbreviations should not be included without explanations in the Abstract. It needs to be thoroughly checked. (Line 27, Line 214, Line 244...)"

Response: Revised (L27-28; L70; L266; L300).

"6) The title for each subsection is a little bit long. Please simplify it in the revision."

[Figure]

Response: The titles have been further simplified.

"7) Section 2.1: The reference or data URL ought to be provided here."

Response: The data URL has been added (L125-127).

"8) Line 101-102: This sentence, which describes data utilization in this work, is better to be placed in Section 2."

Response: This sentence has been placed in Section 2 (L139-140).

"9) Line 203-206: The authors proposed enhanced hygroscopic growth of aerosol due to increase in RH. This point could be further confirmed by the ratios of Lidar-observed aerosol extinction and PM2.5 concentration provided by air quality monitoring network. Thus, more information is suggested to be added here."

Response: If we had obtained lidar observations with aerosol extinction, we would have shown aerosol hygroscopic growth with RH increase more clearly. Currently, relevant lidar observations in Xi'an are exactly what we desire, but beyond the reach. However, aerosol hygroscopic growth with enhanced with increased RH was observed in Beijing, Gucheng in Hebei Province and Lin'an in Zhejiang Province. After moisture absorption in North China, aerosol particle size increases 20%~60% (Pan et al., 2009). Based on our field observations, aerosol scattering coefficient and backscattering coefficient increased by 58 and 25% as the RH increased from 40 to 85 % in Lin'an (Zhang et al., 2015). Aerosol hygroscopic growth caused a 47% increase in the calculated aerosol direct radiative forcing at 85% RH, compared to the forcing at 40% RH. In Gucheng, it is found that the aerosol scattering coefficient and backscattering coefficient increased by 29% and 10%, respectively with RH increasing from 40% to 80% (Qi et al., 2018).

"10) Line 209-201: Regional transport of air pollution from Yuncheng to Xi'an is claimed here. Simple backward trajectories is recommended to be included here to clearly show the transport pathway"

Response: Thanks for the suggestion. We just wanted to point out the inter-regional

transport, which could be approximately shown by the change of near ground wind direction and velocity in different regions without using more complex methods such as back-trajectory.

  Cheng, Y. et al., Reactive nitrogen chemistry in aerosol water as a source of sulfate during haze events in China, Science Advances 2(2016), p. e1601530.

Ding, A.J. et al., Enhanced haze pollution by black carbon in megacities in China, Geophysical Research Letters 43(2016).

Ervens, B., Turpin, B.J., Weber, R.J., Secondary organic aerosol formation in cloud droplets and aqueous particles (aqSOA): a review of laboratory, field and model studies, Atmospheric Chemistry and Physics 11(2011), pp. 11069-11102.

Guo, S. et al., Elucidating severe urban haze formation in China, Proceedings of the National Academy of Sciences of the United States of America 111(2014), pp. 17373-17378.

Jia, Y., Rahn, K.A., He, K., Wen, T., Wang, Y., A novel technique for quantifying the regional component of urban aerosol solely from its sawtooth cycles, Journal of Geophysical Research Atmospheres 113(2008), pp. 6089-6098.

Kuang, Y., Zhao, C.S., Tao, J.C., Bian, Y.X., Ma, N., Impact of aerosol hygroscopic growth on the direct aerosol radiative effect in summer on North China Plain, Atmospheric Environment 147(2016), pp. 224-233.

Liu, X.G. et al., Formation and evolution mechanism of regional haze: a case study in the megacity Beijing, China, Atmospheric Chemistry & Physics 13(2013), pp. 4501-4514.

Pan, X.L. et al., Observational study of influence of aerosol hygroscopic growth on scattering coefficient over rural area near Beijing mega-city, Atmospheric Chemistry & Physics 9(2009), pp. 7519-7530. Pilinis, C., Seinfeld, J.H., Grosjean, D., Water content of atmospheric aerosols, Atmospheric Environment 23(1989), pp. 1601-1606.

Qi, X. et al., Aerosol hygroscopicity during the haze red-alert period in December 2016 at a rural site of the North China Plain, 32(2018), pp. 38-48.

Tie, X. et al., Severe Pollution in China Amplified by Atmospheric Moisture, Scientific Reports 7(2017), p. 15760.

Wang, G. et al., Persistent sulfate formation from London Fog to Chinese haze, Proc Natl Acad Sci U S A 113(2016), pp. 13630–13635.

Wang, H. et al., Mesoscale modeling study of the interactions between aerosols and PBL meteorology during a haze episode in China Jing-Jin-Ji and its near surrounding region - Part 2: Aerosols' radiative feedback effects, Atmospheric Chemistry & Physics 15(2015), pp. 3277-3287.

Zhang, L. et al., Observations of relative humidity effects on aerosol light scattering in the Yangtze River Delta of China, Atmospheric Chemistry & Physics 15(2015), pp. 2853-2904.

Zhong, J. et al., Feedback effects of boundary-layer meteorological factors on cumulative explosive growth of PM2.5 during winter heavy pollution episodes in Beijing from 2013 to 2016, Atmos. Chem. Phys. 18(2018a), pp. 247-258.

Zhong, J. et al., Feedback effects of boundary-layer meteorological factors on explosive growth of PM2.5 during winter heavy pollution episodes in Beijing from 2013 to 2016, Atmospheric Chemistry & Physics(2017a), pp. 1-25.

Zhong, J., Zhang, X., Wang, Y., Liu, C., Dong, Y., Heavy aerosol pollution episodes in winter Beijing enhanced by radiative cooling effects of aerosols, Atmospheric Research 209(2018b), pp. 59-64. Zhong, J. et al., Relative contributions of boundary-layer meteorological factors to the explosive growth of PM 2.5 during the red-alert heavy pollution episodes in Beijing in December 2016, Journal of Meteorological Research 31(2017b), pp. 809-819.

Please also note the supplement to this comment:
https://www.atmos-chem-phys-discuss.net/acp-2018-1077/acp-2018-1077-AC1-supplement.pdf
* * *
[Figure]

[Figure]

**Fig. 1.** The time series of PM2.5 mass concentration in the urban area of Beijing

---

## Author Comment (AC2) · 13 Jan 2019

Authors reply to reviewer's comments:

Dear Anonymous Referees,

Thanks for your careful review of the manuscript. We read the reviewer's comments carefully, and have responded and taken all of reviewer's comments into consideration and revised the manuscript accordingly. My detailed responses, including a point-by-point response to the review and a list of all relevant changes, are as follows:

[Figure]

"Reviewer #2: This paper uses very comprehensive measurements, including PM2.5 mass concentrations, radiosonde observations, etc., to understand the two-way feedback mechanism between unfavorable meteorological conditions and cumulative aerosol pollution in several heavily and less heavily polluted regions in China. Previous studies have been focused on the North China region, while this study can provide a more general picture with extended study regions. "1) My major concern for the paper is that the presentations of the results are descriptive, and I would suggest the authors provide more in-detailed calculations and present in a more quantitative way."

Response: Two figures (Fig.5 and Fig. 16) and corresponding discussions have been added to further quantitatively estimate the magnitude of the two-way feedback from TSs to CSs in different regions (L242-L260; L514-524). To better show the aerosol-induced temperature reduction, we mainly focused on polluted Beijing, Xi'an, and Shenyang, which have more striking meteorological modification.

"2) Near ground temperature bias are shown at different PM2.5 concentrations, however, it is not clear to what extend is contributed by aerosol effects. It is likely that those temperature structures co-occur with different stagnant conditions and thus are different at different PM2.5 concentrations. It would be better to provide more quantitative results here."

Response: The contributions of aerosol-radiation feedback and decrease in turbulent diffusion to PM2.5 growth has been quantified using an online model, GRAPES_CUACE, which also indicates online calculation of the aerosol-radiation feedback is essential for the prediction of PM2.5 explosive growth and peaks during HPEs. With the online aerosol feedback, the modeled local inversion was much closer to the sounding observations (Wang et al., 2018). In this manuscript, we wanted use statistical analysis to show the aerosol-induced temperature modification. Although the effect of aerosols on radiative transfer in the atmosphere is modeled based on prescribed climatological aerosol distributions (Dee et al., 2011), it has not considered the two-way feedback mechanism between the cumulated aerosol pollution and the

worsened meteorological conditions (Simmons, 2006). Therefore, the magnitude of the feedback mechanism could be statistically reflected by the gaps between the ERA-interim reanalysis and the meteorological radiosonde observations. The disparities have been used to present the observational evidence of aerosol-PBL interactions in Beijing (Ding et al., 2016; Huang et al., 2018). Previous studies have shown that HPEs generally included the TSs, whose aerosol pollution formation is primarily caused by pollutants transported from polluted regions, and the CSs, in which the PM2.5 increase is dominated by stable atmospheric stratification characteristic near-ground anomalous inversion, moisture accumulation and reduced BL height under slight or calm winds. During the CSs, the temperature inversion was found to be caused or reinforced mainly by accumulated aerosols. Other factors, including topography, advection and long-wave radiation are likely conducive to weak/normal inversion, but not dominant with respect to anomalous inversion (Zhong et al., 2018a; Zhong et al., 2018b; Zhong et al., 2017).

"3) Line 41-43: It is not surprising that FWRP and YRD regions are largely influenced by inter-regional and trans-regional transport. It would be more valuable to know in these regions, how important are the two-way feedback in cumulative periods, compared to the abrupt injection of transport."

Response: As suggested, we added figures and discussions about the comparison and magnitude of the two-way feedback mechanism in TSs and CSs (Fig. 5 and Fig.16; L242-L260; L514-524), particularly in the polluted Beijing, Xi'an, and Shenyang with more striking meteorological modifications. Overall, the temperature modification was more striking with increasingly worsened aerosol pollution. For different stages, aerosol-induced near-ground cooling bias in the TSs was 18.6%, 48.7% and, 28.2% of that in Beijing, Xi'an and Shenyang. It is expectable because from the TSs to the CSs aerosol pollution worsened with increasing radiative cooling effects. Moreover, though relatively strong winds in the TSs were conducive to pollution transport, they were unfavorable for the formation and maintenance of stable stratification, in which occurred

aerosol self-induced pollution deterioration frequently.

Dee, D.P. et al., The ERA-Interim reanalysis: configuration and performance of the data assimilation system, Quarterly Journal of the Royal Meteorological Society 137(2011), pp. 553-597. Ding, A.J. et al., Enhanced haze pollution by black carbon in megacities in China, Geophysical Research Letters 43(2016).

Huang, X., Wang, Z., Ding, A., Impact of Aerosol-PBL Interaction on Haze Pollution: Multiyear Observational Evidences in North China, Geophysical Research Letters 0(2018). Simmons, A., ERA-Interim: New ECMWF reanalysis products from 1989 onwards, ECMWF newsletter 110(2006), pp. 25-36.

Wang, H. et al., Contributions to the explosive growth of PM 2.5 mass due to aerosol–radiation feedback and decrease in turbulent diffusion during a red alert heavy haze in Beijing–Tianjin–Hebei, China, 18(2018), pp. 17717-17733.

Zhong, J. et al., Feedback effects of boundary-layer meteorological factors on cumulative explosive growth of PM2.5 during winter heavy pollution episodes in Beijing from 2013 to 2016, Atmos. Chem. Phys. 18(2018a), pp. 247-258.

Zhong, J., Zhang, X., Wang, Y., Liu, C., Dong, Y., Heavy aerosol pollution episodes in winter Beijing enhanced by radiative cooling effects of aerosols, Atmospheric Research 209(2018b), pp. 59-64. Zhong, J. et al., Relative contributions of boundary-layer meteorological factors to the explosive growth of PM 2.5 during the red-alert heavy pollution episodes in Beijing in December 2016, Journal of Meteorological Research 31(2017), pp. 809-819.

Please also note the supplement to this comment:
https://www.atmos-chem-phys-discuss.net/acp-2018-1077/acp-2018-1077-AC2-supplement.pdf

---

## Author Response (AR2)

Authors reply to reviewer's comments:

Dear Anonymous Referees,

Thanks for your careful review of the manuscript. We read the reviewer's comments carefully, and have responded and taken all of the reviewer's comments into consideration and revised the manuscript accordingly. My detailed responses are as follows:

> **"Reviewer #1: Most of my comments have been properly addressed and revised in the manuscript. Here are some minor issues that may help further improve this article.**
> **1) Detailed information about UVA measurements needs to be provided."**

**Response:** We have provided more information about UVA observations (L128-136).

> **"2) In some figures like Fig. 8,14,17, the legends obscure some parts of data points. Please revised these figures to make them more clear."**

**Response:** Revised (L339; L480; L569).